# The circulating cell-free DNA landscape in sepsis is dominated by impaired liver clearance

## Graphical abstract

## Authors

Kiki Cano-Gamez, Patrick Maclean, Masato Inoue, ..., Stuart McKechnie, Chun-Xiao Song, Julian C. Knight

## Correspondence

k.e.cano-gamez@exeter.ac.uk (K.C.-G.), julian.knight@well.ox.ac.uk (J.C.K.)

## In brief

Cano-Gamez et al. show that cell-free DNA levels rise sharply during acute sepsis due to impaired liver clearance and that cell-free DNA contains important host and pathogen information, which could improve diagnosis and enhance our understanding of severe infections.

## Highlights

- Cell-free DNA levels in plasma increase over 40-fold during acute sepsis

- This cfDNA buildup is likely caused by impaired liver clearance during disease

- Cell-free DNA retains gene activity clues informative of disease biology

- Pathogen DNA is detectable in the plasma and has diagnostic potential

Cano-Gamez et al., 2025, Cell Genomics 5, 100971
October 8, 2025 © 2025 The Authors. Published by Elsevier Inc.

# Cell Genomics

CellPress

## Article

# The circulating cell-free DNA landscape in sepsis is dominated by impaired liver clearance

Kiki Cano-Gamez,[1,2,7,*] Patrick Maclean,[1] Masato Inoue,[3,4] Sakineh Hussainy,[3,4] Elisabeth Foss,[1] Chloe Wainwright,[1] Hanyu Qin,[1,5] Stuart McKechnie,[6] Chun-Xiao Song,[3,4] and Julian C. Knight[1,5,6,*]

[1]Centre for Human Genetics, Nuffield Department of Medicine, University of Oxford, Oxford, UK
[2]Department of Clinical and Biomedical Sciences, University of Exeter, Exeter, UK
[3]Ludwig Institute for Cancer Research, Nuffield Department of Medicine, University of Oxford, Oxford, UK
[4]Target Discovery Institute, Nuffield Department of Medicine, University of Oxford, Oxford, UK
[5]Chinese Academy of Medical Science Oxford Institute, University of Oxford, Oxford, UK
[6]John Radcliffe Hospital, Oxford University Hospitals NHS Foundation Trust, Oxford, UK
[7]Lead contact
*Correspondence: k.e.cano-gamez@exeter.ac.uk (K.C.-G.), julian.knight@well.ox.ac.uk (J.C.K.)

## SUMMARY

Circulating cell-free DNA (cfDNA) is a promising molecular biomarker, but its role in severe infection is unclear. Here, we profile cfDNA from sepsis patients and controls, demonstrating a 41-fold increase during disease. Methylation-based deconvolution revealed similar cfDNA compositions in the two groups, suggesting that cfDNA accumulation during disease is due not to excess cell death but to impaired hepatic clearance. Fragmentation and end-motif patterns both support this hypothesis, suggesting prolonged exposure of cfDNA to circulating nucleases. In addition, we show that cfDNA retains nucleosome footprints informative of gene activity. By developing a novel method to quantify these footprints and integrate them with single-cell data, we report an increase in cfDNA from Kupffer cells and liver parenchyma in patients with liver dysfunction. Finally, we show that cfDNA contains pathogen-derived material, highlighting its diagnostic potential. This high-throughput, multimodal study provides a reference for understanding cfDNA's role in sepsis and critical illness.

## INTRODUCTION

Circulating cell-free DNA (cfDNA) comprises a collection of DNA fragments that exist outside cells and circulate in the bloodstream. These fragments originate from homeostatic cell death (e.g., apoptosis and efferocytosis) as dying cells release apoptotic bodies and chromatin fragments into the circulation for clearance by the mononuclear phagocyte system.[1] The half-life of cfDNA is estimated to be between minutes and hours,[1] which makes it ideal as a biomarker for real-time monitoring of cell death at remote tissues. Circulating cfDNA has been widely used as a biomarker in pregnancy, organ transplantation, and cancer, where the mixture of genotypes in the cfDNA pool makes it possible to detect pregnancy complications,[2,3] organ rejection,[4–6] and mutated tumor-derived DNA[7–10] before clinical signs of illness are apparent. However, the utility of cfDNA in acute conditions like severe infection remains underexplored.

The rapid and non-invasive nature of cfDNA profiling makes it particularly promising in the context of sepsis, a dysregulated host response to infection associated with life-threatening organ dysfunction.[11,12] Sepsis is one of the leading causes of death worldwide, with 11 million deaths caused by this clinical syndrome every year.[13] Biomarkers to enable earlier detection of organ failure could facilitate clinical decision-making and timely therapeutic intervention, helping address the burden of this devastating condition. In addition, sepsis is a heterogeneous syndrome with a variety of disease mechanisms.[14–20] Consequently, monitoring of cfDNA could help elucidate the mechanisms underlying disease (e.g., by tracking immune cell function or identifying abnormal cell death events), thus improving diagnostic precision and enabling patient stratification.[14]

Previous studies have demonstrated that cfDNA is significantly increased during sepsis and have suggested this relates to disease severity.[21,22] For example, cfDNA concentration in serum is associated with sepsis mortality,[23] and higher levels of circulating mitochondrial DNA (mtDNA) predict poor prognosis.[24] Moreover, the contribution of microbes and infecting pathogens to the cfDNA pool has been extensively studied.[25] However, the characteristics of human-derived cfDNA during sepsis, including its tissues of origin and how it is cleared, are not known.

Recent advances in experimental techniques to assay the epigenome[9,26–31] and computational tools to quantify

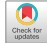

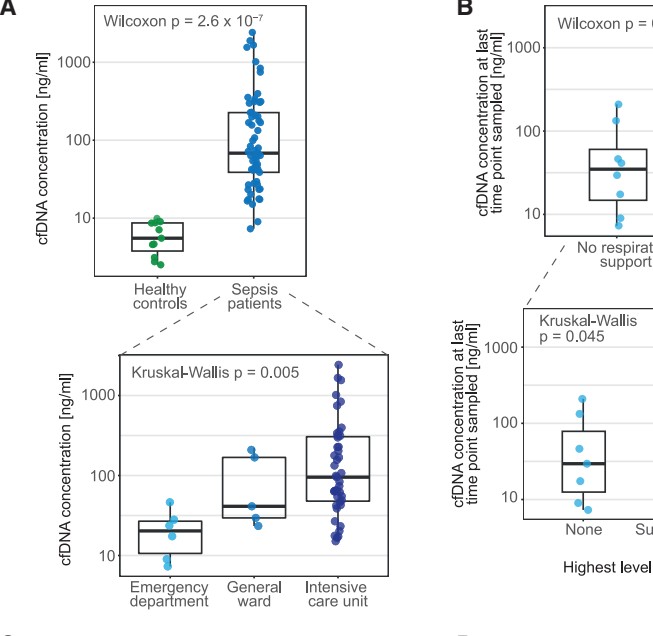

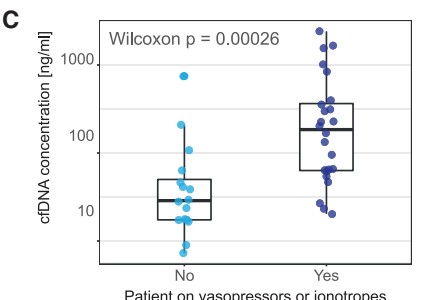

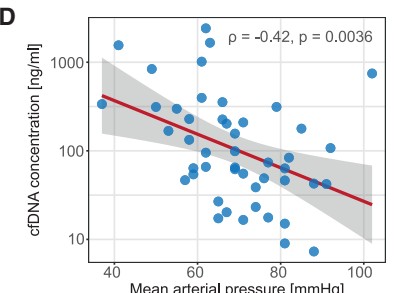

**Figure 1. Levels of cfDNA in circulation reflect sepsis severity**

(A) cfDNA concentration in the plasma of sepsis patients and healthy controls (top) and in sepsis patients within different hospital care settings (bottom). *p* values were derived using a Wilcoxon rank-sum test. Boxplots represent median and interquartile ranges (IQRs) of cfDNA concentration.

(B) cfDNA concentration in sepsis patients stratified by level of respiratory support. Categories are ordered by increasing invasiveness. *p* values were derived using a Wilcoxon rank-sum test. Boxplots represent median and IQRs of cfDNA concentration.

(C) cfDNA concentration in sepsis patients stratified by requirement for treatment with vasopressors or inotropes. *p* values were derived using a Wilcoxon rank-sum test. Boxplots represent median and IQRs of cfDNA concentration.

(D) Correlation between mean arterial pressure (*x* axis) and cfDNA concentration (*y* axis). *p* values and correlation estimates were computed with a Pearson correlation test.

type-specific cell death. Instead, our observations are compatible with cfDNA accumulating due to impaired liver clearance, a hypothesis supported by both fragmentation patterns and end-motif frequencies. Moreover, we show that analysis of methylation and nucleosome positioning in cfDNA could enable development of novel biomarkers for monitoring liver function, organ failure, and immune cell turnover during disease. Finally, we demonstrate that cfDNA contains frag-

cell-type-specific epigenetic contributions[6,32–35] now make it possible to infer which tissues contribute material to the circulation based on the presence of epigenetic marks in cfDNA (e.g., DNA methylation or chromatin modifications). A recent study profiled DNA methylation to understand the composition of cfDNA in several conditions, including sepsis.[6] Observations from this study suggest that sepsis may cause a sharp increase in the amount of neutrophil-derived cfDNA, which supports the known role of neutrophil extracellular traps (NETs) in this disease.[36–38] Moreover, cfDNA composition was shown to reflect instances of organ failure.[6] Despite these advances, an in-depth characterization of the circulating cfDNA landscape (the "circulome") in sepsis is needed to fully understand its clinical potential.

Here, we present a large-scale multimodal study of the circulome during sepsis. By profiling cfDNA methylation, fragmentation patterns, end-motif sequences, and nucleosome positioning across hospitalized sepsis patients, we present an in-depth characterization of the tissues of origin of cfDNA and identify potential mechanisms driving cfDNA accumulation. Our observations confirm that cfDNA increases during acute sepsis. However, we observe no systematic differences in cfDNA composition between sepsis patients and controls, suggesting that cfDNA accumulation is not primarily driven by cell-

ments of microbial origin and show how this information could be used to aid diagnosis.

## RESULTS

### Levels of cfDNA in circulation reflect sepsis severity

To assess the utility of cfDNA in sepsis, we isolated cfDNA from 86 plasma samples from 46 patients admitted to hospital with sepsis and serially sampled as part of the Sepsis Immunomics study, together with 12 healthy controls (STAR Methods). We observed that cfDNA concentration was dramatically higher in sepsis patients compared to healthy controls, with an average 41.2-fold increase (Figure 1A). When stratifying patients by their required level of care, we found a significant increase in cfDNA in patients requiring higher care levels, with patients in the intensive care unit (ICU) showing concentrations approximately 10-fold higher than patients in the emergency department (ED) or general medical ward (Figure 1A).

To confirm the association between cfDNA concentration and illness severity, we stratified patients by their required level of respiratory support, as well as by the presence of cardiovascular dysfunction requiring treatment (e.g., hypotension or shock). cfDNA concentration increased gradually for patients requiring progressively more invasive respiratory support (Figure 1B),

**A**

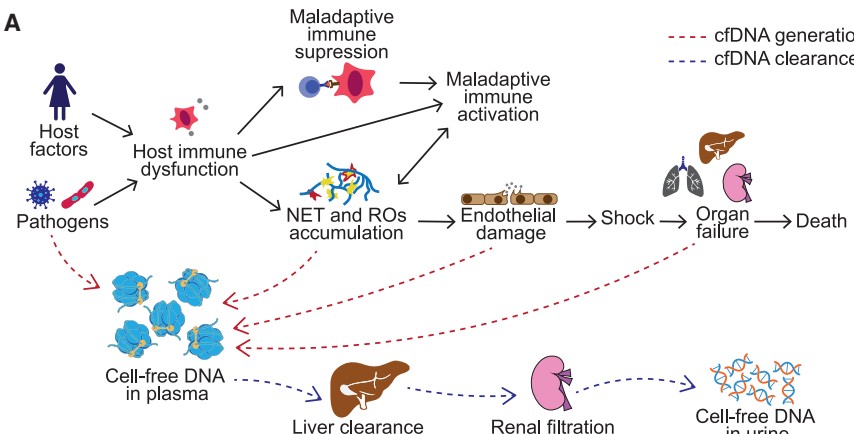

**B**

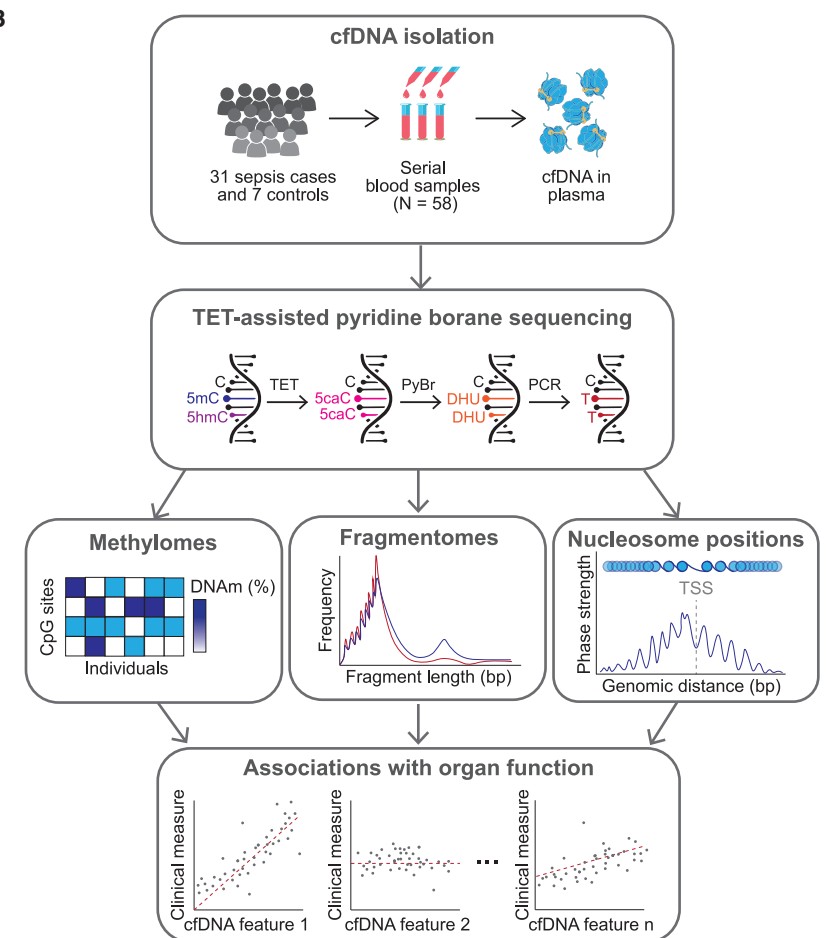

**Figure 2. An experimental approach for multimodal profiling of circulating cfDNA**
(A) Schematic of possible mechanisms contributing and clearing DNA from the circulation. Solid arrows represent putative causal relationships between disease processes. Dashed red and blue arrows represent possible mechanisms of cfDNA production and clearance during sepsis, respectively.
(B) Experimental approach followed in this study. Blood samples were collected from sepsis patients and used for cfDNA isolation. cfDNA was subsequently used for library preparation and sequencing with the TAPS protocol, which enabled simultaneous characterization of its methylation status, its fragmentation patterns, and evidence of nucleosome positioning. This information was then related back to clinical measures of illness severity and used to infer the likely tissues of origin of cfDNA.

that cfDNA levels are reflective of illness severity, motivating further study on the biology of this molecular compartment.

### An approach for multimodal cfDNA profiling

We next asked why cfDNA increases during sepsis. Based on our current understanding of sepsis,[12] we proposed a set of mechanisms that could explain the observed cfDNA accumulation. We grouped these into two: (1) mechanisms that increase cfDNA release and (2) mechanisms that reduce cfDNA clearance (Figure 2A). The first group comprised cell death concomitant with the host immune response (e.g., NETosis, pyroptosis, or necrosis), damage of the vascular endothelium, and organ failure. In contrast, the second group included hepatic clearance and renal filtration.[26] We reasoned that the relative contributions of these mechanisms could be distinguished based on cfDNA composition, which motivated us to leverage cfDNA methylation, fragmentation, and nucleosome positioning to identify the likely tissues of origin of cfDNA.

We used ten-eleven translocation (TET) enzyme-assisted pyridine borane sequencing (TAPS), a method that enables reliable genome-wide DNA methylation profiling.[9,26] In contrast to bisulfite sequencing (BS-seq)[39,40] and enzymatic methyl sequencing (EM-seq),[28] TAPS is ideally suited to low DNA input, and its direct conversion of methylated cytosines does not reduce sequence diversity, enabling efficient bioinformatic analysis of sequence-based DNA features. We used TAPS to profile DNA methylation

suggesting it is related to the extent of respiratory failure. Moreover, cfDNA levels were significantly higher in patients requiring vasopressor or inotrope treatment (Figure 1C). This suggested a relationship between hypotension and cfDNA, as confirmed by a significant negative correlation between cfDNA concentration and mean arterial pressure (MAP) on the day of sampling (Figure 1D). Taken together, these observations demonstrate

**Table 1. Clinical and demographic information on our cfDNA TAPS cohort**

|  | Healthy volunteers | Sepsis patients |
|---|---|---|
| Sample size | | |
| Number of participants | 7 | 33 |
| Number of samples | 7 | 51 |
| Median samples per participant | 1 | 1 (range: 1–4) |
| Age | | |
| Mean | 43 | 61 |
| 95% CI | (37, 48) | (58, 63) |
| Self-reported sex | | |
| Male (%) | 57 | 69 |
| Female (%) | 43 | 31 |
| Sepsis source | | |
| Community-acquired pneumonia (CAP) | N/A | 7 (21%) |
| Urosepsis | N/A | 6 (18%) |
| Fecal peritonitis (FP) | N/A | 4 (12%) |
| Biliary sepsis | N/A | 3 (9%) |
| Necrotizing fasciitis | N/A | 3 (9%) |
| Bacterial meningitis | N/A | 2 (9%) |
| Abdominal sepsis | N/A | 1 (3%) |
| Infective colitis | N/A | 1 (3%) |
| Infective endocarditis | N/A | 1 (3%) |
| Leptospirosis | N/A | 1 (3%) |
| Septic arthritis | N/A | 1 (3%) |
| Unknown | N/A | 3 (9%) |
| Participant location | | |
| Emergency department (ED) | N/A | 4 (12%) |
| General ward | N/A | 2 (6%) |
| Intensive care unit (ICU) | N/A | 27 (82%) |

in 58 cfDNA samples from 31 patients and 7 healthy controls, of which 56 passed our quality filters (STAR Methods). The demographic and clinical characteristics of this cohort are summarized in Table 1 and Figure S1. We leveraged these data to study (1) the cfDNA methylome, (2) the cfDNA fragmentation landscape (i.e., fragment size distributions and end-motif frequencies), and (3) nucleosome positioning at gene regions (Figure 2B). We used these complementary data modalities to infer the likely tissues of origin of cfDNA and to assess its utility as a biomarker for sepsis monitoring.

### Variation in cfDNA methylation reflects disease processes

We first assessed the quality of our methylation data. Based on the performance of our spike-in controls, we confirmed that TAPS could successfully identify methylated CpG sites (i.e., 5mC + 5hmC) at 93% sensitivity with only a 0.35% false-positive rate (Figure 3A). This gave us confidence that our data were high quality. We proceeded to analyze a set of 19,288,064 autosomal CpG sites reliably detected across samples (STAR Methods). To maximize power, we collapsed CpGs into units based on

genomic proximity by segmenting the genome into non-overlapping 1-kb tiles (STAR Methods). In agreement with previous literature,[41] this revealed two types of genomic regions: those with average methylation of approximately 0% (hypomethylated) and those with mean methylation of 75% (hypermethylated). Highly variable regions were predominantly hypermethylated (Figure 3B).

We next asked if methylation variance was concentrated in any specific genomic elements. While lowly variable regions segregated randomly across annotations, the top 10% most variable regions were overwhelmingly enriched in introns, with a smaller proportion localizing to exons and promoters (Figure 3C). In contrast, variable regions were depleted from CpG islands (CpGIs), which are known to be hypomethylated and invariant.[41–44] The high level of variance in introns supports recent observations suggesting gene activity correlates most strongly with DNA methylation at the first intron.[45] Given that introns accounted for most of the observed variation, we aggregated CpG sites located within the first intronic region of all known genes (STAR Methods). In contrast with promoters, which showed both hyper- and hypomethylation, first introns showed consistently high methylation proportions and higher variability (Figure 3D). We used this intron set as the basis for exploratory analysis.

We next asked which factors shape the cfDNA methylome during sepsis. We used variance partitioning analysis[46] to identify the factors explaining most of the variance (STAR Methods). Surprisingly, only a small proportion of variance (mean = 1.55%) was explained by disease status, with most of the variation being either unexplained or accounted for by interindividual variability (Figure 3E). This could reflect a high contribution of unmeasured factors (e.g., genetic background and infection history) as has been reported for cellular DNA methylation.[47,48] It could also relate to disease heterogeneity. We thus repeated this analysis on sepsis patients only, leveraging information from their medical records to quantify the contribution of clinical factors (STAR Methods). While a large proportion of variance remained unexplained, we observed a clear influence of organ function on cfDNA methylation. For example, 1,932 regions showed >30% of methylation variance explained by levels of alanine aminotransferase (ALT), a marker of liver dysfunction[49–51] (Figure 3E). Similar observations were made for creatinine, a marker of kidney dysfunction,[52] and C-reactive protein (CRP) and the neutrophil-to-lymphocyte ratio (NLR), markers of systemic inflammation[53–55] (Figure 3E). This suggests that fluctuations in organ function result in cfDNA methylation changes at selected regions. Time since ICU admission also had a sizable impact (Figure 3E), suggesting that some methylation changes reflect disease trajectories.

Given its high contribution to methylation variance, we next focused on the impact of liver dysfunction. Principal-component analysis (PCA) revealed a clear separation of patients with high ALT measurements from the remaining cohort (Figure 3F). In contrast, disease status did not correlate with principal components (Figure 3F). The distribution of samples in PCA space demonstrated a higher level of variability within the patient group than between patients and controls, a phenomenon that could relate to the wide variability in severity of organ dysfunction

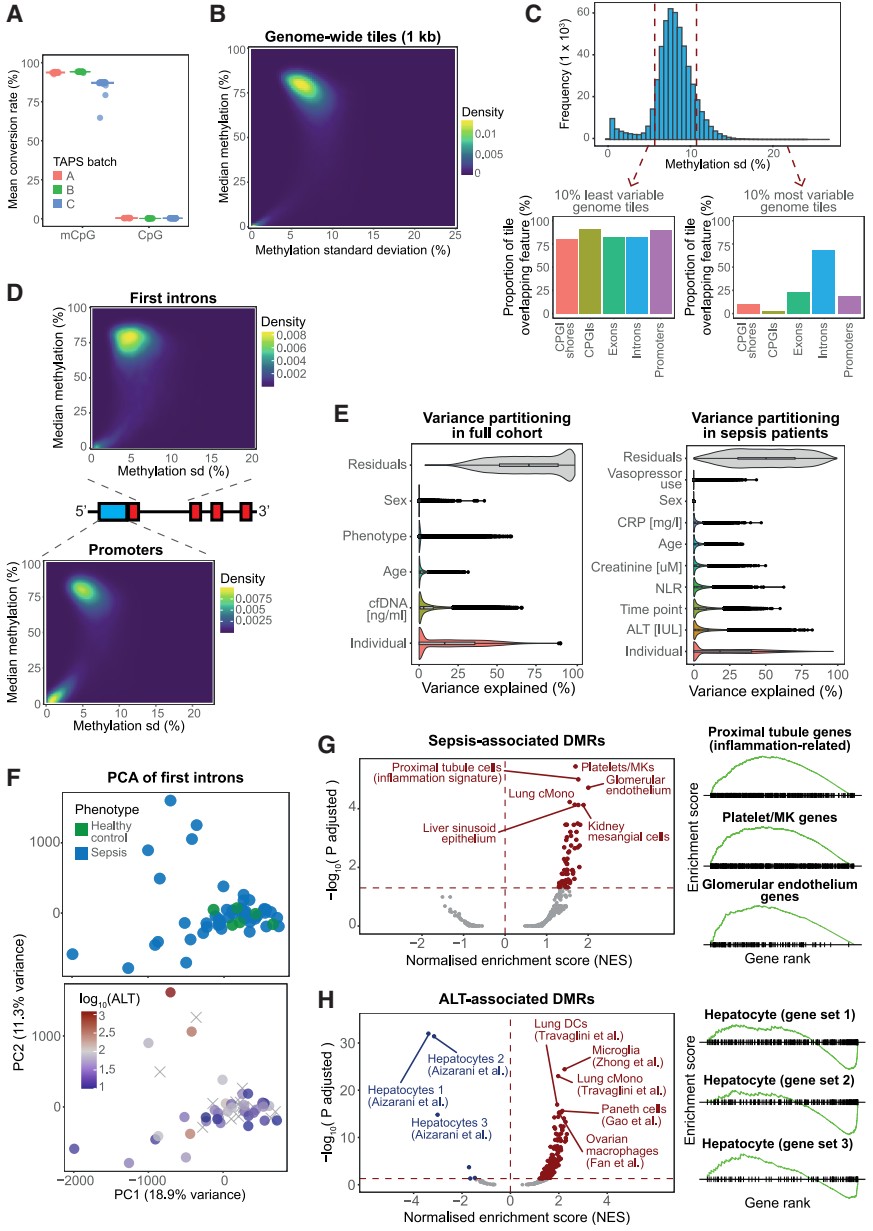

**Figure 3. Variation in the cfDNA methylome reflects organ function and disease processes**

(A) mCpG conversion rates in positive and negative spike-in controls as estimated based on our full cohort. Each dot represents a sample, with colors indicating library preparation batch.

(B) Two-dimensional-density plot showing the variability (standard deviation, *x* axis) and average proportion (%, *y* axis) of cfDNA methylation at 1-kb genomic windows as estimated based on our full cohort. Shades of color are proportional to the number of genomic regions located within a given range.

(C) Histogram of cfDNA methylation variability (standard deviation) across all 1-kb windows in the genome (top) as estimated based on our full cohort. The proportional overlap between 1-kb windows and known genomic annotations is shown for the top 10% most variable and the bottom 10% least variable genomic windows (bottom).

(D) Two-dimensional-density plots showing the variability and average cfDNA methylation at first intron (top) and promoter (bottom) regions as estimated based on our full cohort. Shades of color are proportional to the number of regions located within a given range.

(E) The proportion of methylation variance explained by different variables was estimated at intronic regions using variance partitioning analysis. This analysis was conducted separately for the full cohort (left) and for sepsis patients only (right). Violin plots show the distribution of variance explained by each variable, with dots representing estimates for individual introns.

(F) Principal-component analysis based on methylation at first introns. Dots represent samples, with colors indicating either disease status (top) or ALT concentration (bottom). Crosses indicate samples for which ALT measurements were not available.

(G) Volcano plot showing GSEA-derived enrichment scores (*x* axis) and FDR-adjusted *p* values (*y* axis) for genomic regions ranked by strength of association between cfDNA methylation and disease status (left). Enrichment was estimated based on our full cohort. A subset of gene sets is highlighted. Enrichment score distributions for the topmost enriched gene sets are also shown (right).

(H) Volcano plot showing GSEA-derived enrichment scores (*x* axis) and FDR-adjusted *p* values (*y* axis) for genomic regions ranked by strength of association between cfDNA methylation and ALT levels (left). Enrichment was estimated based on sepsis samples only. A subset of gene sets is highlighted. Enrichment score distributions for the topmost enriched gene sets are also shown (right).

within sepsis. To test this, we performed differential methylation and gene set enrichment analyses (GSEAs) (STAR Methods). We identified 4,208 genomic regions where DNA methylation was associated with ALT levels (ALT-associated DMRs) at 5% false discovery rate (FDR). Conversely, only eight regions were differentially methylated between sepsis and controls (sepsis-associated DMRs). Sepsis-associated DMRs were enriched in gene sets related to platelets, epithelium, and inflammation (Figure 3G). In contrast, ALT-associated DMRs showed a striking and significant negative enrichment in hepatocyte-specific

genes (Figure 3H). Since DNA methylation and gene activity are negatively correlated,[41,45] this is reflective of a higher hepatocyte contribution to cfDNA. Thus, our observations suggest that patients with high ALT have a significantly higher proportion of hepatocyte-derived DNA in circulation.

Taken together, our observations suggest that the circulating methylome is substantially more variable within sepsis patients than it is between patients and controls and that a subset of genomic regions shows methylation patterns that reflect organ dysfunction and disease trajectories.

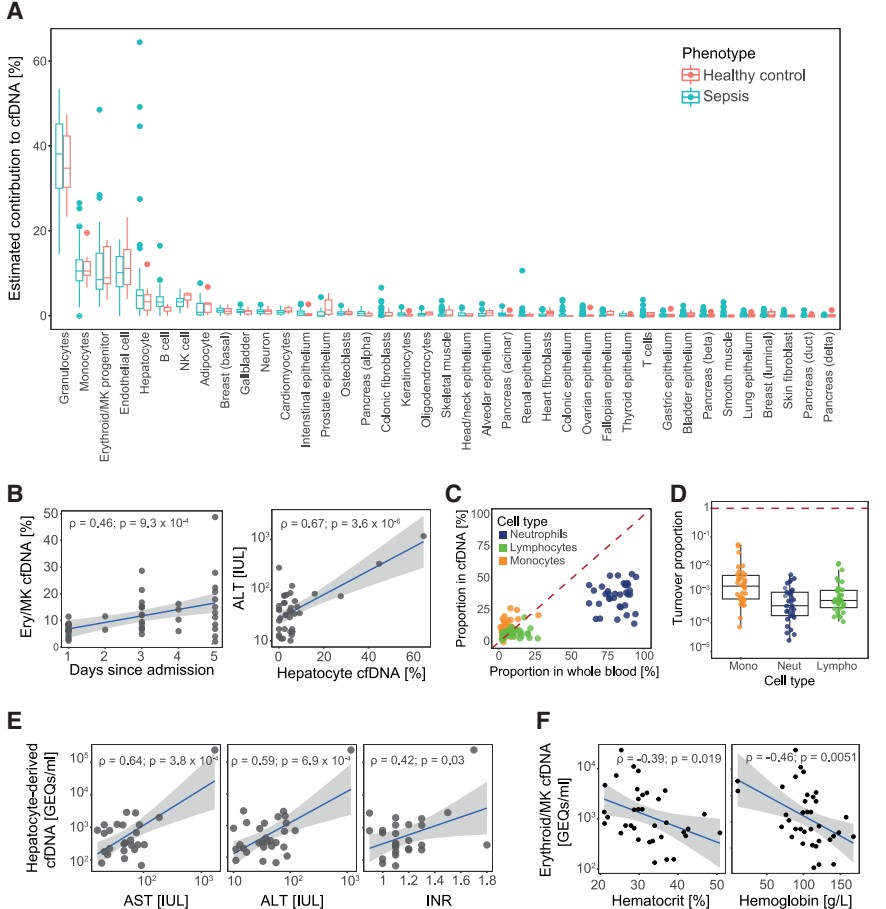

**Figure 4. Analysis of tissues of origin of cfDNA during sepsis**

(A) Proportion of cfDNA estimated to arise from different tissues based on methylome deconvolution. Boxplots show median and interquartile ranges (IQRs) of estimated proportions in sepsis patients (red; $n$ = 49 samples from 31 patients) and healthy controls (blue; $n$ = 7 independent samples).

(B) Correlations between time since hospital admission and proportion of MEP-derived cfDNA (left), as well as ALT levels and proportion of hepatocyte-derived cfDNA (right). Correlation coefficients and $p$ values were estimated using Pearson correlation tests.

(C) Correlation between proportions of cells in circulation ($x$ axis) and their cellular contributions to cfDNA ($y$ axis). Colors indicate the three main immune cell lineages. The identity line is shown for reference.

(D) Cellular turnover rates estimated based on cfDNA GEQs and circulating cell proportions. Boxplots represent median and IQRs of cellular turnover proportions. The dashed red line indicates a turnover rate of 1, corresponding to cell death and cell generation being equal.

(E) Correlation between markers of liver dysfunction ($x$ axis) and hepatocyte-derived cfDNA GEQs ($y$ axis). Blue lines and shaded regions indicate linear fits and confidence intervals. Correlation coefficients and $p$ values were estimated using Pearson correlation tests.

(F) Correlation between clinical markers of erythropoiesis ($x$ axis) and MEP-derived cfDNA GEQs ($y$ axis). Blue lines and shaded regions indicate linear fits and confidence intervals. Correlation coefficients and $p$ values were estimated using Pearson correlation tests.

## cfDNA accumulation in sepsis is not cell-type specific

We next leveraged methylation profiles to infer the relative contributions of different tissues to cfDNA. We used EpiDISH, a methylome deconvolution method,[56] to estimate tissue-of-origin information based on a BS-seq reference tissue atlas previously developed for cfDNA deconvolution[32] (STAR Methods). Granulocytes showed the highest contribution to cfDNA (37.2% on average), followed by monocytes (11.4%), megakaryocyte-erythroid progenitors (MEPs; 11.3%), lymphocytes, and a much smaller contribution from solid organs (Figure 4A). In contrast to other tissues, hepatocytes and endothelial cells contributed substantially more DNA to the circulation. These observations agree with previous studies, which have identified erythropoiesis and neutrophil turnover as the two main sources of cfDNA.[6,32,57]

We next compared cfDNA composition between patients and controls. Surprisingly, we observed no overall difference in composition, with cell-type contributions ranked in the same order regardless of disease status (Figure 4A). These results are in stark contrast with previous literature suggesting an increase in granulocyte-derived cfDNA during sepsis[6] and point to accumulation of cfDNA in sepsis as not being primarily due to increased immune cell death.

To ensure these results were not biased by technical limitations, we performed further testing. First, we applied our deconvolution approach to a subset of samples of known cellular composition. We isolated whole blood leukocytes (WBLs) from 15 sepsis patients and 5 healthy controls and performed DNA methylation profiling (STAR Methods). We used these samples to compare cellular proportions estimated from DNA methylation to those directly measured in hospital using a complete blood count (CBC) (STAR Methods). In contrast to cfDNA, the WBL methylome was overwhelmingly dominated by cellular composition (Figures S2A and S2B), and estimated cell proportions were highly concordant between methylome deconvolution and direct measurements (Figures S2C–S2F). Moreover, deconvolution successfully distinguished WBL samples from positive controls containing purified neutrophil DNA only (Figure S2). These observations greatly strengthened our confidence in our approach.

To ensure our conclusions were not biased by our choice of deconvolution reference, we repeated our analysis using a different tissue atlas[6] (STAR Methods). Results from both atlases were highly concordant, with neutrophils, MEPs, and monocytes remaining the top contributors to the cfDNA pool and cell proportions being largely independent of disease status (Figure S3). Thus, we concluded that the observed lack of compositional

differences between cfDNA from patients and controls was unlikely to reflect technical biases. These observations are incompatible with cfDNA accumulation being a result of higher cell-type-specific DNA release (e.g., increased endothelial damage, NETosis, or pyroptosis) and instead suggest a potential impairment in cfDNA clearance.

### Variation in cfDNA composition between patients reflects disease processes

We next turned our attention to the relationship between cfDNA and disease features. While overall cfDNA composition did not differ by disease status, we noticed a much larger amount of variation within sepsis patients than in the control group. For example, sepsis patients showed a much broader range of hepatocyte-derived cfDNA proportions, with hepatocytes contributing as much as 60% of DNA in some patients (Figure 4A). We observed similarly large variability in MEPs, monocytes, and neutrophils. This could reflect disease heterogeneity.

To investigate this, we tested for associations between cfDNA composition and clinical variables (STAR Methods). We observed that the proportion of MEP-derived cfDNA was low after hospital admission but increased over the course of hospitalization, gradually returning to a value closer to the proportion in healthy controls (Figure 4B). This could reflect hematopoietic rewiring during the early stages of sepsis, a well-known phenomenon that occurs during infection to support higher production of effector immune cells at the expense of erythrocytes and other hematopoietic lineages.[58] Similarly, the proportion of hepatocyte-derived cfDNA was significantly associated with clinical measures of liver dysfunction (Figure 4B), suggesting a higher contribution of this cell type in individuals with liver damage. These observations supported our hypothesis that variation in the contributions of different tissues to the cfDNA pool reflects disease features.

We next integrated our deconvolution results with cfDNA concentration measurements to estimate the number of genomes contributed by each cell type (genome equivalents, GEQs)[6] (STAR Methods). GEQs are proportional to the number of cell death events occurring within a given cell type, thus providing a principled method to track cellular turnover. We first focused on circulating immune cells. We observed little correlation between the proportion of immune cells in circulation (derived from clinical CBCs) and their proportion in cfDNA (Figure 4C). This is well documented[59] and supports the theory that cfDNA composition is proportional to cell death, not to the number of live cells present at any given time. We then estimated cellular turnover rates as the ratio of cell death events (cfDNA GEQs) to live cells (CBC counts) per unit volume of blood (STAR Methods). Across all major immune lineages, the number of live cells greatly exceeded the number of cell death events. This was particularly evident for neutrophils, which showed the lowest turnover rates (Figure 4D). These observations agree with the role of emergency granulopoiesis in sepsis, where neutrophil production is greatly increased to support the host immune response.[58,60]

We then turned our attention to liver dysfunction. We confirmed that hepatocyte-derived cfDNA GEQs were significantly associated with clinical signs of liver dysfunction, including ALT, aspartate aminotransferase (AST), and the international normalized ratio (INR), a measure of the synthetic function of the liver (Figure 4E). Thus, hepatocyte-derived cfDNA correlates with liver dysfunction. Similarly, we found a significant association between MEP-derived cfDNA GEQs and clinical measures of erythroid production (hematocrit and hemoglobin levels). This implies that MEP-derived cfDNA correlates with hematopoietic rewiring and fluctuations in red blood cell (RBC) production.

Taken together, these observations suggest that tracking cfDNA composition is a promising strategy for real-time assessment of disease processes during sepsis.

### cfDNA fragmentation reveals impaired hepatic clearance during sepsis

Our observations suggest that cfDNA clearance may be impaired during sepsis. cfDNA is cleared by Kupffer cells in the liver, a process referred to as hepatic clearance.[61] DNA and chromatin fragments released from hepatic clearance are then removed from the bloodstream by the kidney (renal filtration), resulting in their excretion as ultra-short fragments in the urine.[62] Thus, we next asked if hepatic clearance was impaired during sepsis.

Circulating cfDNA exists as small chromatin fragments, which originate from apoptosis.[63] These fragments, one or two nucleosomes in size, are cleaved by circulating nucleases like deoxyribonuclease I (DNASE1) and deoxyribonuclease I-like 3 (DNASE1L3).[64,65] Thus, we reasoned that if hepatic clearance were impaired, cfDNA would remain in the circulation longer, therefore being exposed to nucleases for a prolonged period. We hypothesized that this would leave a distinct fragmentation signature, which could be detected using sequencing.

We used read length information to infer the distribution of cfDNA fragment sizes (STAR Methods). In agreement with previous literature,[66] most cfDNA fragments (mean = 90.75%, median = 91.9%, SD = 3.25%) were mononucleosomal particles (120–220 bp; mode = 165 bp), with a smaller subset formed of two (mean = 5.2%; 220–420 bp; mode = 343 bp) or more (0.61%; >420 bp; mode = 524 bp) nucleosomes (Figure 5A). Mononucleosomal fragments showed ~10 bp periodicity, a common feature of cfDNA that arises from the orientation of DNA grooves around nucleosomes during cleavage.[67] This gave us confidence in the quality of our fragmentation information.

We next asked if sepsis changed cfDNA fragmentation. Cell-free DNA was significantly more fragmented in sepsis, with an enrichment in mononucleosomes and a depletion of larger fragments (Figure 5B). To describe this skew, we used a previously proposed approach[63,64,68] to calculate cfDNA fragmentation indices (FIs), defined as the ratio of mononucleosomal to dinucleosomal fragments (STAR Methods). FIs were significantly increased in sepsis (Figure 5C), consistent with cfDNA being more fragmented. We also noted that FIs decreased over the course of hospitalization (Figure 5D). This suggests that cfDNA features may exhibit complex temporal dynamics.

To understand if the observed skew in fragmentation was driven by impaired hepatic clearance, we tested for associations between cfDNA fragmentation and liver function tests. FIs were significantly correlated with ALT, AST, and INR, as would be

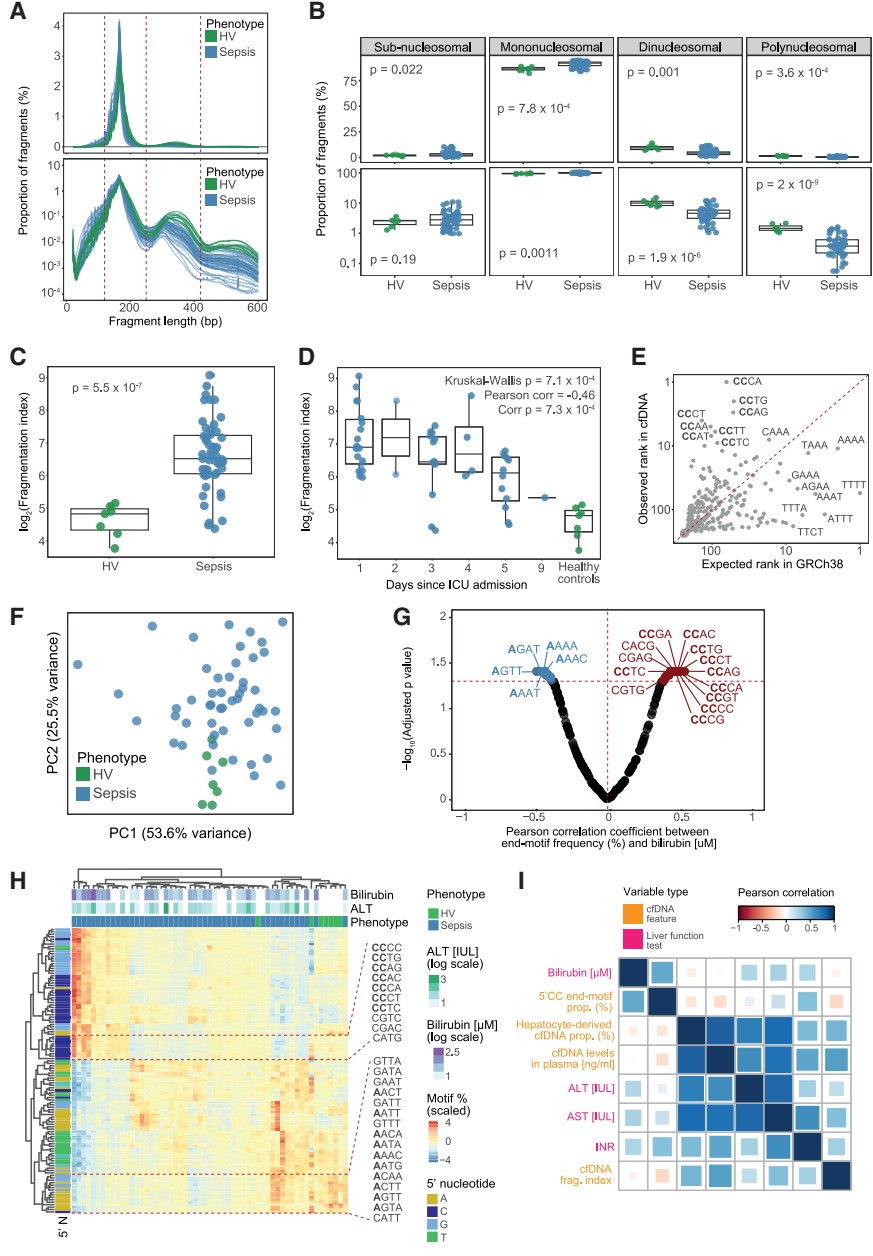

**Figure 5. cfDNA fragmentation reveals impaired hepatic clearance during sepsis**

(A) The estimated percentage of cfDNA fragments (y axis) of different lengths (x axis) in linear (top) and logarithmic (bottom) scales. Each line represents a sample, with colors indicating control and sepsis patient samples ($n$ = 51 samples from 31 patients and 7 control samples).

(B) Percentage of cfDNA fragments classified as subnucleosomal, mononucleosomal, dinucleosomal, or polynucleosomal based on their observed length. Proportions are shown in both linear (top) and logarithmic (bottom) scales. Each dot represents a sample, with colors indicating disease status. Boxplots show median and IQR values for each sample group. $p$ values were estimated using Wilcoxon rank-sum tests.

(C) cfDNA fragmentation indices (y axis, in logarithmic scale) stratified by disease status (x axis). Each dot represents a sample, with colors indicating disease status. Boxplots show median and IQR values for each sample group. $p$ values were estimated using Wilcoxon rank-sum tests.

(D) cfDNA fragmentation indices (y axis, in logarithmic scale) stratified by time since admission (x axis). Each dot represents a sample, with colors indicating disease status. Boxplots show median and IQR values for each group. $p$ values were estimated using a Kruskal-Wallis test. For sepsis samples, correlation coefficients and $p$ values between fragmentation indices and time were computed using a Pearson correlation test and are also shown.

(E) Comparison of average observed (y axis) and expected (x axis) cfDNA end-motif frequencies in our cohort. Each dot represents a 4-bp 5′ end motif, with a subset of motifs off the diagonal being highlighted. The identity line is shown for reference.

(F) Principal-component analysis plot based on the frequencies of all 4-bp 5′ end motifs. Each dot represents a sample, with colors indicating disease status.

(G) Volcano plot showing the correlation between end-motif frequency and bilirubin. Each dot represents a motif, with its correlation coefficient (x axis) and FDR-adjusted $p$ value (y axis) from Pearson correlation tests shown. Significantly positively and negatively correlated motifs are highlighted in red and blue, respectively.

(H) Heatmap of 5′ cfDNA end-motif frequencies across all samples in our cohort. Shades of color represent end-motif frequencies, with marginal color bars indicating the first nucleotide in the motif (horizontal axis) as well as disease status and liver-specific sequential organ failure assessment (SOFA) scores (vertical axis). Samples were grouped using hierarchical clustering of motif frequencies. A subset of motifs showing appreciable differences in frequency is highlighted.

(I) Correlation plot between all pairwise combinations of liver function tests and cfDNA features. Each square indicates results from a pairwise correlation test, with shades of color indicating the estimated Pearson correlation coefficient. Variable names are shown, with colors indicating whether they were derived from liver function testing or from cfDNA sequencing. Squares were grouped using hierarchical clustering.

expected from impaired hepatic clearance (Figure S4A). The same was true for cfDNA concentration (Figure S4A). This agrees with previous literature,[69] supporting the prominent role of the liver in shaping the cfDNA landscape.

Nucleases show biases in their cutting site preferences.[64] Thus, we hypothesized that impaired clearance should also

leave behind a distinctive end-motif signature in cfDNA, reflecting prolonged exposure to circulating nucleases. To test this, we leveraged our data to infer fragment end-motif sequences and estimate their proportion (STAR Methods). cfDNA end motifs were significantly enriched in 5′-CCNN sequences, known to be preferentially generated by DNASE1L3, compared to their

expected frequency based on the composition of the human genome (Figure 5E). This agrees with the known role of DNASE1L3 in cfDNA generation.[63,64,68]

We next assessed the impact of sepsis on the end-motif landscape. PCA of end-motif frequencies showed a clear separation between patients and controls (Figure 5F). To understand whether the observed difference in end-motif frequencies reflected fluctuations in hepatic clearance, we tested for associations between end-motif usage and liver function tests (STAR Methods). We observed that 5′-CC end motifs, associated with DNASE1L3, were more frequent in sepsis patients and that they were positively correlated with bilirubin (Figures 5G and 5H). In contrast, 5′-A motifs, associated with the activity of the nuclease DNA Fragmentation Factor Subunit Beta (DFFB), were observed at lower frequencies in sepsis patients and were negatively correlated with bilirubin (Figures 5G and 5H). While DNASE1L3 is present in the circulation, DFFB is an intracellular nuclease known to play a key role in apoptosis.[63,64] Thus, our observations are compatible with cfDNA being originally derived from apoptosis but impaired hepatic clearance subsequently causing prolonged exposure of cfDNA to DNASE1L3 in the bloodstream.

While the correlation between end motifs and bilirubin supports this hypothesis, we were surprised to find no associations between end-motif frequencies and AST or ALT, the two most widely used markers of hepatic injury. To better understand this, we retested for associations between all cfDNA features and liver function tests. This revealed that cfDNA concentration in plasma and FIs correlated with ALT, AST, and NLR, but not with bilirubin (Figure S4A). The opposite was true for end-motif frequency (i.e., 5′-CC motifs), which was significantly associated with bilirubin and INR, but not with AST or ALT (Figure S4A). This suggests that some cfDNA features reflect hepatocyte damage (i.e., a hepatocellular pattern of liver injury), while others may reflect dysfunction in the biliary system (i.e., a cholestatic pattern of liver injury).[70]

Previous studies have reported that some critically ill patients progress from an initial hepatocellular injury (often caused by impaired perfusion) to a later cholestatic injury, marked by elevation in bilirubin.[71,72] These patients show a delay of about 3 days between peak ALT and peak bilirubin.[71] Thus, we reasoned our observations could be driven by different clinical trajectories, with cfDNA features exhibiting complex temporal profiles. To test this, we assessed the behavior of liver function tests and cfDNA features over time. Both ALT and AST peaked at day 1 post-ICU admission in our cohort (Figure S4B), matching the temporal trajectory of FIs (Figure 5D). In contrast, bilirubin peaked at day 5 (Figure S4B), with this late increase driven by only a small subset of patients.

To further explore this, we computed correlations between all pairwise combinations of features (Figure 5I). This revealed three groups: (1) variables that reflect hepatocellular injury (ALT, AST, cfDNA concentration, percentage of hepatocyte-derived cfDNA, and FIs), (2) variables that reflect cholestatic injury (bilirubin and the proportion of 5′-CC cfDNA end motifs), and (3) variables altered in both injury types (INR). This demonstrates how different features of cfDNA capture different mechanisms of liver

dysfunction. This is encouraging, as it showcases how multiple orthogonal layers of information can be extracted from the same biomarker.

## cfDNA retains cell-type-specific nucleosome signatures

Recent studies have demonstrated that cfDNA retains nucleosome footprints reflective of chromatin architecture in its tissues of origin.[73–76] This holds great promise for cfDNA-based biomarkers, as methylation profiling requires lengthy experimental protocols that cannot be easily adapted to clinical settings. In contrast, nucleosome footprinting can be inferred using more widely available DNA sequencing. This motivated us to map nucleosome footprints, assess whether they retain tissue specificity, and compare them to our observations from methylome deconvolution.

We adapted an approach by Snyder et al.[73] to estimate windowed protection scores (WPSs) around the transcriptional start site (TSS) of known genes (STAR Methods). WPSs are designed to capture the degree of DNA protection from cleavage by nucleases. Because nucleases predominantly cut DNA at the linker region between nucleosomes, WPSs reflect nucleosome positioning.[73] Analysis of the WPS signal revealed a clear nucleosome footprint at gene regions (Figure 6A). This footprint was characterized by a dip in signal at the TSS, known to be a nucleosome-free region (NFR), and subsequent periodic oscillations, which peaked approximately every 180 bp, consistent with the known length of DNA wrapped around nucleosomes (Figure 6A).

Nucleosome positioning at active genes is characterized by the first nucleosome (+1 nucleosome) being consistently placed at approximately the same position across cells, a phenomenon known as phasing.[77–79] The strength of nucleosome phasing decreases for nucleosomes positioned farther away from the TSS, which eventually return to a random arrangement. These features were well captured in our data, with WPS values and oscillation amplitudes weakening proportionally to their distance from the TSS (Figure 6A).

Having confirmed that our data retain nucleosome footprints, we turned our attention to the impact of disease status. The strength of nucleosome phasing was substantially higher in patients compared to controls (Figure 6A). This could be driven by higher levels of gene activity in sepsis, where immune cells and tissues are known to activate a variety of gene expression programs.[15,37,80,81] To confirm that nucleosome footprints were reflective of gene activity, we stratified genes based on their expression level. We used data from the Genotype-Tissue Expression (GTEx) consortium to define sets of inactive and active genes (STAR Methods). Active genes showed a distinct periodic WPS signal, which was sharply increased in sepsis (Figure 6B). In contrast, inactive genes showed no difference in WPS between patients and controls and contained no evidence of nucleosome phasing, as demonstrated by the absence of periodic oscillations (Figure 6B). We next asked if genes expressed at different levels showed appreciable differences in their nucleosome footprints. We used data from the UK Genomic Advances in Sepsis (GAinS) study[15,20,80] to classify genes into four groups based on their level of expression in whole blood during sepsis (STAR Methods). Footprint strength increased progressively

**Figure 6. Cell-free DNA retains cell-type-specific nucleosome positioning signatures**

(A) Average WPS values and their associated 95% confidence intervals were estimated for all known gene regions using a 5-kb window centered at the TSS. Blue and red lines indicate estimates for sepsis patients and healthy controls, respectively ($n$ = 51 samples from 31 patients and 7 control samples). Dotted vertical lines indicate the expected position of each nucleosome and are spaced 187 bp from one another. A regional plot focusing on a 2-kb region around the TSS is also shown (top).

(B) Average WPS values at the TSS regions of active (right) and inactive genes (left), based on gene expression measurements from the GTEx study. Colors indicate disease status, and dotted vertical lines indicate the expected position of each nucleosome.

(C) Average WPS values at the TSS regions of increasingly more active genes, based on gene expression measurements from the GAinS study. Colors indicate disease status.

(D) Observed WPS values at the TSS regions of all known genes (solid lines) are shown alongside predictions derived from a dampened harmonic oscillator model (dotted lines). Colors indicate disease status. The equation used for model fitting is shown for reference.

(E) Dampened harmonic oscillator models were fitted separately to each sample in our study. The coefficient of variation of estimates for each model parameter is shown as a bar plot.

(F) Heatmap of correlation estimates computed between all model parameter estimates. Colors indicate Pearson correlation coefficients. Rows and columns were ordered by similarity using hierarchical clustering.

(G) Dampened harmonic oscillator models were fitted to the TSS region of tissue-specific gene sets. Bar plots show the Pearson correlation coefficient ($y$ axis) between each model parameter estimate and the proportion of hepatocyte-derived cfDNA as estimated based on DNA methylation. Each plot shows estimates for a different model parameter, with colors indicating the gene set used during model fitting.

(H) Volcano plot of correlations between hepatocyte-derived cfDNA proportions estimated from methylome deconvolution and WPS exponential decay rates estimated for different liver cell types. The dotted line indicates the statistical significance threshold of Benjamini-Hochberg (BH) adjusted $p$ < 0.05. Colors indicate the gene set used during model fitting.

from genes in the bottom quartile to genes in the top gene expression quartile (Figure 6C), demonstrating a positive relationship between gene activity and nucleosome phasing. Taking these results together, we confirmed that nucleosome footprints are detectable in cfDNA and that they are indicative of gene activity.

We next sought to develop an analytical method to quantitatively study nucleosome phasing. Previous studies have approached this problem using methods like Fourier series analysis, which can approximate the WPS signal using a series of trigonometric functions.[73,75] However, results from these approaches are often difficult to interpret. Instead, we reasoned

that nucleosome positioning could be modeled by combining an exponential decay function with a dampened harmonic oscillator, a model widely used in classical physics.[82] In brief, we assume that nucleosome phasing strength decays over genomic distance, that the nucleosome footprint signal oscillates periodically, and that the amplitude of these oscillations decays proportional to their distance from the TSS (STAR Methods). We used non-linear least-squares analysis to fit this model to WPS values across our cohort (STAR Methods). Comparison of the observed WPSs to the predictions from this approach confirmed that this model adequately captures the oscillations in WPS, as well as the decay in phasing over distance (Figure 6D).

We next fitted models separately to each sample in our study (STAR Methods). Model fitting performed reliably across the sample set, with high concordance between observed and predicted WPS values (Figure S4). We asked to what extent different nucleosome phasing parameters varied across samples. Parameters describing the oscillation frequency (i.e., phase shift, period, and angular frequency) showed little variability, as indicated by their coefficients of variation (CVs) being close to 0. These parameters relate to nucleosome size, which is fixed and invariant (Figure 6E). Reassuringly, the median oscillation period inferred by our models was 182 bp (median = 182 bp, mean = 183.1 bp, SD = 8.52 bp), which matches the known length of DNA around each nucleosome. In contrast, parameters describing nucleosome phasing (i.e., oscillation amplitude and exponential decay rate), as well as the decay in phasing over distance (i.e., oscillation decay rate), showed much higher levels of variability, with CVs >0.6 (Figure 6E).

We asked if any of these parameters captured important aspects of cfDNA biology and whether they supported our observations from cfDNA methylation. Visual inspection of the cross-correlation between model parameters revealed three parameter groups: (1) those related to nucleosome size and spacing (angular frequency and phase shift), (2) those related to signal strength at the TSS (oscillation amplitude and exponential decay rate), and (3) those related to nucleosome phasing downstream of the TSS (oscillation decay rate) (Figure 6F). Parameters in the second group were consistently and significantly increased in sepsis (Figure S5), concordant with a generalized increase in gene activity. Thus, we reasoned that parameters describing WPS signal strength at the TSS were the most likely to retain tissue-specific information.

To test this, we repeated our analysis using tissue-specific gene sets from single-cell studies (STAR Methods). Given the importance of liver function, we focused on liver-specific cell types. We collated gene sets active in liver cell types from the liver atlas of Aizarani et al.[83] These gene sets comprised both core structural cell types (i.e., hepatocytes, liver sinusoidal endothelial cells [LSECs], and microvascular endothelial cells [MVECs]), as well as transient and resident immune cells (Kupffer cells, resident B cells, and natural killer [NK] and natural killer T [NKT] cells). We fitted separate models to each gene set and assessed the correlation of the inferred parameters with methylome-derived hepatocyte proportions (STAR Methods). This enabled us to identify parameters that capture tissue-specific gene activity and assess the agreement between DNA methylation and nucleosome positioning. While most parameters showed little evidence of correlation, we observed consistently positive and high correlation between methylation-derived hepatocyte proportions and nucleosome phasing exponential decay rates across most tested cell types (Figure 6G). This confirmed our hypothesis that signal strength at the TSS is reflective of tissue-specific gene activity and that this parameter correlates well with estimates derived from DNA methylation. Further inspection revealed statistical evidence of correlation for core liver cell types (hepatocytes, MVECs, and LSECs) but non-significant correlations across most immune cells (resident B cells and NKT cells). Kupffer cells, the resident liver macrophages in charge of degrading

cfDNA, were a clear exception, showing the highest correlations (Figure 6H).

To confirm this, we visually inspected nucleosome footprints for each gene set, stratifying samples based on their levels of ALT. This analysis confirmed that higher ALT is accompanied by a more pronounced nucleosomal footprint in core liver cell types and Kupffer cells but not in other liver immune cells (Figure S6). These observations demonstrate that nucleosome positioning retains tissue-specific information, which can be leveraged to estimate the tissues of origin of cfDNA. Moreover, they suggest that liver damage secondary to sepsis results in an increase in cfDNA derived from the liver parenchyma and Kupffer cells.

Taken together, the results showed that cfDNA data retain nucleosome footprints and that these contain tissue-specific information. We developed a principled approach to quantitatively study these footprints and proposed that parameters that relate to footprint strength at the TSS of cell-type-specific genes can be used to estimate the contribution of these cell types to cfDNA.

## cfDNA contains information on infecting pathogens

Previous research has shown that cfDNA comprises a small proportion of fragments of microbial origin.[25,84,85] Microbial DNA in circulation is promising in sepsis, where infection is the root cause, yet identification of the pathogen causing the infection is often challenging. It has been suggested that metagenomic analysis of circulating cfDNA could help circumvent this limitation as well as enabling simultaneous profiling of the pathogen and the host immune response.[86] Motivated by this, we asked whether cfDNA contained any information about infecting pathogens.

We retrieved sequencing reads that did not map to the human genome and classified them into microbial taxa using a database of all assembled bacterial, viral, archaean, and fungal genomes (STAR Methods). We then quantified microbial reads and defined microbial abundances as the proportion of non-human reads assigned to each taxon (STAR Methods). Metagenomic analyses are highly sensitive to contamination.[87] However, we reasoned that, while microbial genera detected in healthy volunteers were probably contaminants, those observed in sepsis patients were more likely to be disease relevant. Thus, we tested for differential microbial abundance between patients and controls and asked if microbes with higher abundance in the control group were enriched in reported metagenomic contaminants.[25,87] Our analysis revealed a clear enrichment, confirming that microbes detected in healthy controls are likely contamination (Figure 7A). This increased our confidence that, conversely, microbial DNA in septic patients may represent true infecting pathogens.

We next focused on sepsis samples and asked if the microbial composition reflected known patient characteristics. PCA of microbial abundances revealed two main sources of variation: the first component separated a sample with unusually high levels of *Sphingopyxis* DNA, while the second component separated two samples with high levels of *Escherichia* DNA (Figure 7B). The patient with high *Sphingopyxis* abundance showed no detectable growth in blood cultures, and while this genus has been found in polluted water, there is little evidence of it being hazardous to human health.[88] This suggests that this signal may reflect

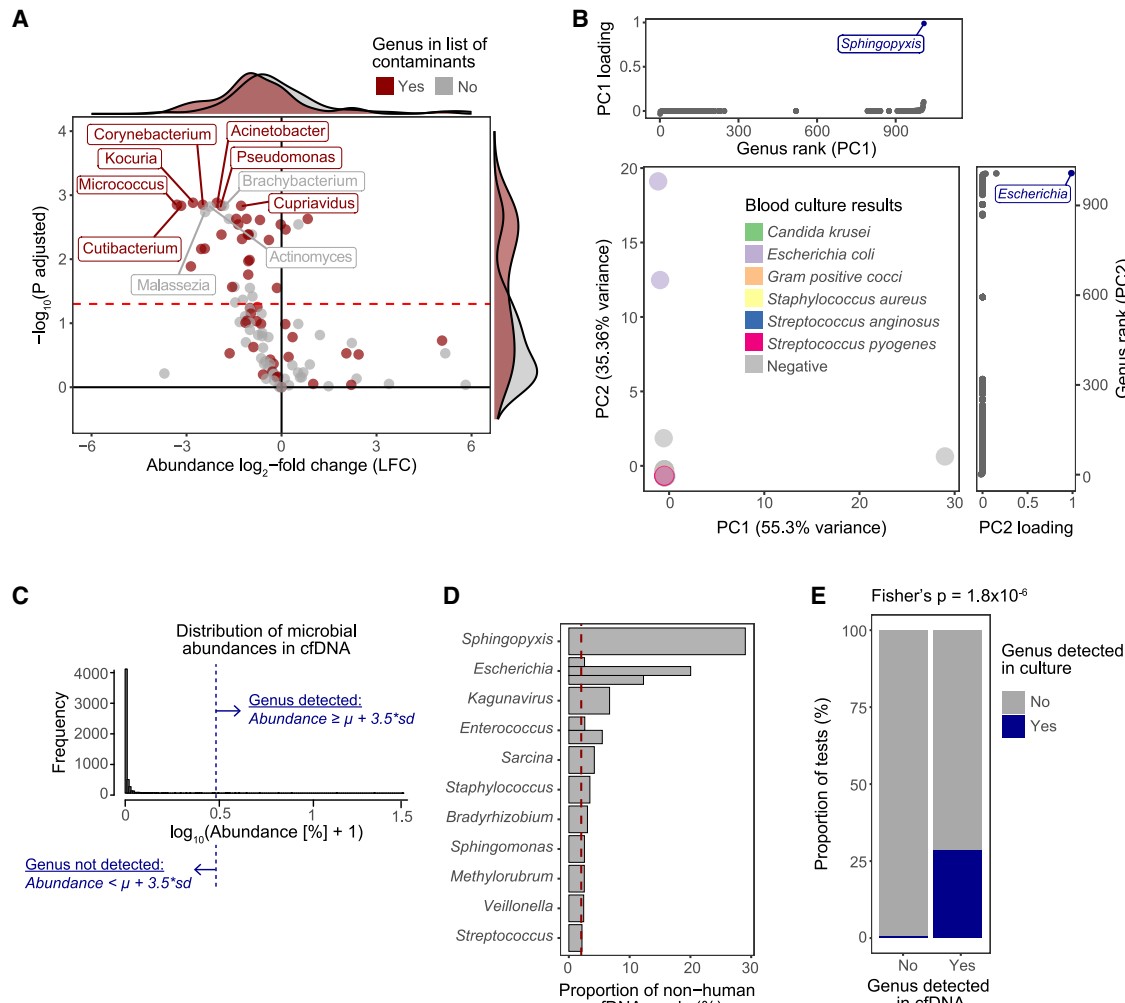

**Figure 7. Cell-free DNA contains information on infecting pathogens**

(A) Volcano plot showing microbial genera that are differentially abundant between sepsis patients and healthy controls. Microbial abundance in circulation (*x* axis) and statistical evidence (*y* axis) are shown, with each dot representing a microbial genus. Red dots indicate known contaminants reported in previous metagenomic studies. Marginal density plots indicate the distribution of known contaminant species.

(B) Principal-component analysis was performed on a matrix of microbial abundances from sepsis patient samples. Each dot represents a sample, with colors highlighting microbiological results from blood cultures. Marginal plots show the contribution of different genera to each principal component.

(C) Histogram showing the distribution of microbial abundances in cfDNA. The dotted line indicates the position of an outlier filter, which we used as a threshold to separate true signals from unspecific background.

(D) Bar plot of abundances for the 11 genera passing our outlier filter threshold. Each bar represents a sample that tested positive for the genus in question, with abundance estimates shown on the *x* axis.

(E) Stacked bar plot showing the proportions of microbial genera detected in cfDNA that are also detected in blood or bodily fluid cultures. These proportions were compared to those of the cfDNA background using a Fisher's exact test.

sample contamination. In contrast, both patients with detectable *Escherichia* DNA in circulation showed positive blood cultures for *Escherichia coli*.

We next set out to distinguish disease-relevant signals from a large background of microbial genera with low read counts. We applied a stringent outlier filter, which we defined as signals with an abundance at least 3.5 SD above the cohort mean (Figure 7C). Only 11 microbial genera passed this threshold, including the two samples with positive *E. coli* cultures, a sample with *Staphylococcus* DNA and positive cultures for *Staphylococcus aureus*,

and two fecal peritonitis samples with detectable *Enterococcus* DNA (Figure 7D). Finally, we asked whether the 11 outlying microbial genera detected were also observed in microbiological cultures. We found that 28.6% of these genera were confirmed by microbiological testing (Figure 7D). This proportion was significantly higher than that of background (i.e., non-outlying) microbial genera, which were seen in cultures only 0.6% of the time.

In summary, we demonstrated that cfDNA contains disease-relevant information about the infecting pathogens underlying

sepsis. While this is encouraging, further studies with larger sample sizes and deeper microbiological characterization will be needed to reliably assess the value of cfDNA in pathogen detection.

## DISCUSSION

Circulating cfDNA is a promising biomarker. However, its utility in sepsis has not been fully explored. Here, we presented a multimodal study of cfDNA during sepsis, profiling methylomes, fragmentation landscapes, and nucleosome footprints.

Our observations demonstrate accumulation of cfDNA during disease and show that the levels of cfDNA reflect sepsis severity, in agreement with previous literature.[89–91] Moreover, we leveraged DNA methylation and computational deconvolution[34,92] to infer the contributions of different tissues to cfDNA and to study cellular turnover,[59] demonstrating that cfDNA composition reflects disease processes like hematopoietic rewiring. Surprisingly, we found no evidence of cfDNA accumulation being driven by increased immune cell death (e.g., NETosis or pyroptosis). Instead, we hypothesize that the observed increase in cfDNA in the absence of compositional differences is likely explained by impaired hepatic clearance during sepsis. This is supported by increased cfDNA fragmentation and higher frequencies of DNASE1L3-associated end motifs, both of which suggest a prolonged exposure of cfDNA to nucleases that is directly proportional to biomarkers of liver dysfunction. It is worth noting how this differs from a recent study of cfDNA in a healthy population, where individuals with higher levels of cfDNA showed lower frequencies of DNASE1L3 motifs.[69] These contrasting observations suggest that different mechanisms of DNA accumulation operate in health and disease.

The observed role of liver-mediated cfDNA clearance is in line with recent *in vivo* studies, which showed that therapeutic agents that block hepatic clearance increase the amount of DNA recovered from liquid biopsies.[93] This therapeutic blockade is reminiscent of our observations in sepsis. Moreover, our observations also agree with a study of cfDNA in healthy individuals, which reported an association between liver function tests and cfDNA concentration.[69] Taken together, these results indicate that cfDNA levels are tightly regulated by the liver, both during homeostasis and in acute infection.

In addition, we demonstrated that cfDNA retains nucleosome footprints and that these contain cell-type-specific gene activity information. While this has been shown before,[73,75] here, we presented a principled approach to study nucleosome phasing and infer cell-type-specific information that relies on an exact mathematical model. When integrating our approach with data from single-cell studies, we observed that sepsis patients with liver dysfunction have a higher proportion of cfDNA derived from the liver parenchyma and Kupffer cells.

Finally, we demonstrated the value of microbial-derived cfDNA. We showed that cfDNA from patients retains information on pathogens, which is often confirmed by microbiological cultures. However, we also observed microbial genera with high contributions to cfDNA but with no detectable growth. It is well known that a large proportion of infecting pathogens do not grow in laboratory settings, with some sepsis patients having detectable microbial DNA and a corresponding increase in mor-

tality yet no observable microbial growth.[94] This could explain why some of our signals were not confirmed by microbiological testing. For instance, both samples with detectable *Enterococcus* DNA came from the same patient sampled at different times. This patient presented with fecal peritonitis, a condition where leakage of *Enterococci* from the gut is common. This suggests that the signal is reproducibly detected over time, unlikely to be a contaminant, and consistent with disease biology, even in the absence of positive microbiology. This is encouraging and motivates future studies on cfDNA-based pathogen detection.

### Limitations of the study

Our study suffers from several limitations. First, our sample size is low, and thus, we are constrained to detect only large effects. Moreover, our control group is formed of healthy individuals, which makes it challenging to understand which alterations are specific to sepsis and which are driven by hospitalization. Future studies with better suited control groups (e.g., critically ill patients without infection) will be instrumental to clarify this.

Second, it is unclear which mechanisms are responsible for impaired cfDNA clearance. Candidate mechanisms may include Kupffer cell death, Kupffer cells being co-opted to fight infection, reduced blood flow due to shock, or occlusion of hepatic blood flow by immune complexes or NETs. Functional and imaging studies will be needed to clarify this.

Third, it is unclear how cfDNA information regarding liver function could be harnessed for biomarker discovery. On the one hand, cfDNA-based biomarkers would benefit from (1) cfDNA's short half-life compared with aminotransferases; (2) cfDNA being amenable to measurement by qPCR, with a very low limit of detection; (3) future improvements to deconvolution techniques making it possible to pinpoint damage in specific anatomical locations of the liver; and (4) a single test distinguishing between multiple types of liver dysfunction. On the other hand, it is unclear how these features change over time, whether they precede liver dysfunction, and how much value they would add to the already existing battery of sensitive liver function tests.

Fourth, it is not known how much of the cfDNA landscape is genetically controlled. Population-wide studies will be needed to map the genetic architecture of cfDNA features like fragmentation, end-motif frequency, and methylation.

Fifth, TAPS remains too lengthy for use in clinical settings. New technologies that enable direct methylation profiling with minimal sample preparation (e.g., Oxford Nanopore) could help reduce turnaround times.[57,95–100] Moreover, these technologies could also detect epigenetic marks such as 5hmC and 6mA, which are more cell-type specific.[101–103]

Finally, our experimental protocols are not ideal for microbial DNA profiling. Techniques specifically developed for damaged DNA (e.g., single-stranded library preparation methods) could help us bridge this gap.[104]

## RESOURCE AVAILABILITY

### Lead contact

Further information and requests for resources should be directed to and will be fulfilled by the lead contact, Kiki Cano-Gamez (k.e.cano-gamez@exeter.ac.uk).

### Materials availability

This study did not generate new unique reagents.

### Data and code availability

The codes used for processing and analysis of cfDNA TAPS data are publicly available on GitHub (https://github.com/jknightlab/TAPS-pipeline/), with a permanent release of this pipeline (v.1.0.0) also deposited in Zenodo (https://doi.org/10.5281/zenodo.15974773).

TAPS-derived cfDNA methylation matrices containing estimated methylation proportions at 19,288,064 autosomal and variable CpG sites, as well as their associated metadata, are publicly available on Zenodo (https://doi.org/10.5281/zenodo.14844792). Raw sequencing files in FASTQ format for the data presented in this study are available via the European Genome-Phenome Archive (EGA; study ID: EGAS50000001033, dataset ID: EGAD50000001505) under controlled access.

### ACKNOWLEDGMENTS

We thank all the patients, patients' families, nurses, and clinicians who participated in the Sepsis Immunomics and GAinS studies. This work was funded, in whole or in part, by the Medical Research Council (MR/V002503/1) (J.C.K.), Wellcome Trust Investigator Award (204969/Z/16/Z) (J.C.K.), Wellcome Trust grants (090532/Z/09/Z and 203141/Z/16/Z) to core facilities Centre for Human Genetics, Chinese Academy of Medical Sciences (CAMS) Innovation Fund for Medical Science (CIFMS), China (grant number 2018-I2M-2-002), Ludwig Institute for Cancer Research (C.-X.S.), and NIHR Oxford Biomedical Research Centre (J.C.K. and C.-X.S.). The views expressed are those of the authors and not necessarily those of the NHS, the NIHR, or the Department of Health. For the purpose of open access, the authors will apply a CCBY public copyright license to any Author Accepted Manuscript version arising from this submission.

### AUTHOR CONTRIBUTIONS

Conceptualization, K.C.-G. and J.C.K.; supervision, J.C.K. and C.-X.S.; data analysis, K.C.-G., P.M., and E.F.; interpretation of results, K.C.-G., P.M., M.I., and J.C.K.; experimental work, K.C.-G., P.M., M.I., S.H., E.F., and C.W.; patient recruitment, J.C.K., H.Q., and S.M.; writing, K.C.-G., P.M., and J.C.K.

### DECLARATION OF INTERESTS

J.C.K. reports a grant to his institution from the Danaher Beacon Programme for work on RNA biomarker point-of-care test development in sepsis for endotype assignment, which includes support for K.C.-G., C.W., and J.C.K.

### STAR★METHODS

Detailed methods are provided in the online version of this paper and include the following:

- KEY RESOURCES TABLE
- EXPERIMENTAL MODEL AND STUDY PARTICIPANTS
  - Sepsis Immunomics cohort
  - Healthy volunteer cohort
- METHOD DETAILS
  - Collection and processing of plasma samples
  - Collection and processing of whole blood leukocytes and CD66$^+$ cells
  - Cell-free DNA isolation from plasma
  - DNA isolation from whole blood leukocytes
  - Preparation of TAPS spike-in controls
  - TAPS library preparation
  - Targeted EM-seq library preparation
  - DNA sequencing
- QUANTIFICATION AND STATISTICAL ANALYSIS
  - Read alignment and filtering of TAPS data
  - Methylation calling from TAPS data
  - CpG quality filtering and summarisation of TAPS methylation data
  - Methylation exploratory analysis and batch correction (TAPS data)
  - Variance partitioning analysis of TAPS data
  - Differential methylation and pathway enrichment analyses of TAPS data
  - Deconvolution of cfDNA tissues of origin from TAPS data
  - Cellular turnover analysis
  - Analysis of cfDNA fragmentation patterns
  - Analysis of cfDNA fragment end-motif frequencies
  - Cell-free DNA-based nucleosome footprinting
  - Analysis of nucleosome footprints for different gene sets
  - Modelling of nucleosome phasing using a harmonic oscillator
  - Analysis of microbial DNA in circulation
  - Adapter trimming, read alignment, and methylation calling of EM-seq data
  - CpG quality filtering of EM-seq data
  - Variance partitioning analysis of EM-seq data
  - Deconvolution of whole blood leukocyte methylomes from EM-seq data

### SUPPLEMENTAL INFORMATION

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

## STAR★METHODS

### KEY RESOURCES TABLE

| REAGENT or RESOURCE | SOURCE | IDENTIFIER |
|---|---|---|
| **Chemicals, peptides, and recombinant proteins** | | |
| Recombinant mouse Tet1 catalytic domain (mTet1CD) | Liu et al.[26] | N/A (produced in-house) |
| Pyridine borane (PyBr) | ThermoFisher Scientific (Alfa Aesar) | Cat# L13178.09 |
| Buffer EB | Qiagen | Cat# 19086 |
| HEPES buffer | Sigma Aldrich | Cat# H3375 |
| **Critical commercial assays** | | |
| EasySep™ HLA chimerism whole blood CD66b positive selection cocktail | StemCell Technologies | Cat# 178882A |
| EasySep™ dextran RapidSpheres | StemCell Technologies | Cat# 178882A |
| EasySep™ red blood cell lysis buffer | StemCell Technologies | Cat# 178882A |
| QIAamp™ circulating nucleic acid kit: Buffer ACL | Qiagen | Cat# 55114 |
| QIAamp™ circulating nucleic acid kit: Buffer ACB | Qiagen | Cat# 55114 |
| QIAamp™ circulating nucleic acid kit: Buffer ACW1 | Qiagen | Cat# 55114 |
| QIAamp™ circulating nucleic acid kit: Buffer ACW2 | Qiagen | Cat# 55114 |
| QIAamp™ circulating nucleic acid kit: Buffer AVE | Qiagen | Cat# 55114 |
| QIAamp™ circulating nucleic acid kit: Qiagen proteinase K | Qiagen | Cat# 55114 |
| Qubit™ dsDNA HS reagent | ThermoFisher Scientific | Cat# Q32854 |
| Qubit™ dsDNA HS buffer | ThermoFisher Scientific | Cat# Q32854 |
| Qubit™ 1X dsDNA HS standard #1 | ThermoFisher Scientific | Cat# Q32854 |
| Qubit™ 1X dsDNA HS standard #2 | ThermoFisher Scientific | Cat# Q32854 |
| Cell-free DNA ScreenTape | Agilent | Cat# 5067-5630 |
| Cell-free DNA sample buffer | Agilent | Cat# 5067-5633 |
| Cell-free DNA ladder | Agilent | Cat# 5067-5632 |
| Monarch™ gDNA blood lysis buffer | New England Biolabs | Cat# T3013-1 |
| Monarch™ gDNA binding buffer | New England Biolabs | Cat# T3014-1 |
| Monarch™ gDNA wash buffer | New England Biolabs | Cat# T3015-1 |
| Monarch™ gDNA elution buffer | New England Biolabs | Cat# T3016-1 |
| Monarch™ RNase A | New England Biolabs | Cat# T3018-1 |
| Proteinase K, molecular biology grade | New England Biolabs | Cat# P8200AAVIAL |
| Phusion™ high-fidelity PCR master mix with HF buffer | New England Biolabs | Cat# M0531S |
| NEBNext Ultra™ II end-repair/dA-tailing module | New England Biolabs | Cat# E7546L |
| KAPA HyperPrep kit | Roche | Cat# 07962363001 |
| KAPA HiFi HotStart Uracil+ ReadyMix kit | Roche | Cat# 07959079001 |
| AMPure XP beads | Beckman Coulter | Cat# A63881 |
| NEBNext™ Enzymatic Methyl-seq kit | New England Biolabs | Cat# E7120L |
| Twist Human Methylome Panel | Twist Bioscience | Cat# 105521 |
| **Deposited data** | | |
| cfDNA TAPS raw sequencing data | This study | EGA study number: EGAS50000001033 EGA dataset ID: EGAD50000001505 |
| cfDNA TAPS methylation calls | This study | Zenodo: https://doi.org/10.5281/zenodo.14844791 |
| GAinS study RNA-seq data | Cano-Gamez et al.[20] | EGA: EGAD00001008730 |
| GTEx v10 summary expression data | GTEx Portal[105] | https://gtexportal.org/home/downloads/adult-gtex/overview |

*Continued*

| REAGENT or RESOURCE | SOURCE | IDENTIFIER |
|---|---|---|
| **Oligonucleotides** | | |
| IDT primers for PCR amplification (5' to 3'): Forward: AATGATACGGCGACCACCGAGATCTACAC Reverse: CAAGCAGAAGACGGCATACGAGAT | Siejka-Zielińska et al.[9] | N/A |
| **Recombinant DNA** | | |
| pNIC28-Bsa4 | Addgene | Cat# 26103 |
| CpG umethylated 2-kb control DNA | Liu et al.[26] | N/A (produced in-house) |
| CpG methylated lambda control DNA | Liu et al.[26] | N/A (produced in-house) |
| Control DNA CpG methylated pUC19 | New England Biolabs | Cat# E7122AAVIAL |
| Control DNA CpG unmethylated Lambda | New England Biolabs | Cat# E7123AAVIAL |
| **Software and algorithms** | | |
| TAPS data processing pipeline (v1.0.0) | This study | https://doi.org/10.5281/zenodo.15974773 |
| TrimGalore (v0.6.10) | Babraham Institute | https://www.bioinformatics.babraham.ac.uk/projects/trim_galore/ |
| BWA MEM (v0.7.17) | Li and Durbin[106] | https://bio-bwa.sourceforge.net |
| SAMtools (v1.8) | Li et al.[107] | http://www.htslib.org/ |
| Picard (v2.23) | Broad Institute | https://broadinstitute.github.io/picard/ |
| MethylDackel (v0.6.1) | GitHub | https://github.com/dpryan79/MethylDackel |
| BEDtools (v2.31) | BEDtools | https://bedtools.readthedocs.io/en/latest/ |
| BBMap | BBTools | https://github.com/BioInfoTools/BBMap |
| twoBitToFa | UCSC Genome Browser | https://genome.ucsc.edu/goldenpath/help/twoBit.html |
| Kraken2 | Wood et al.[108] | https://ccb.jhu.edu/software/kraken2/ |
| nf-core's methyl-seq data analysis pipeline (v2.3.0) | nf-core[109] | https://nf-co.re/methylseq/2.6.0/ |
| Bismark (v0.22.3) | Krueger et al.[110] | https://www.bioinformatics.babraham.ac.uk/projects/bismark/ |
| R (v4.3.2) | The R project for statistical computing | https://www.r-project.org/ |
| R package: methylkit (v1.28.0) | Akalin et al.[111] | https://doi.org/10.18129/B9.bioc.methylKit |
| R package: ggplot2 (v2.3.4.4) | Tidyverse | https://ggplot2.tidyverse.org/ |
| R package: SVA (v3.50) | Zhang et al.[112] | https://doi.org/10.18129/B9.bioc.sva |
| R package: variancePartition (v1.32.5) | Hoffman et al.[46] | https://doi.org/10.18129/B9.bioc.variancePartition |
| R package: limma (v3.58.1) | Ritchie et al.[113] | https://doi.org/10.18129/B9.bioc.limma |
| R package: fGSEA (v1.28) | Korotkevich et al.[114] | https://doi.org/10.18129/B9.bioc.fgsea |
| R package: EpiDISH (v2.18.0) | Zheng et al.[56] | https://doi.org/10.18129/B9.bioc.EpiDISH |
| R package: tidyverse (v2.0.0) | Tidyverse | https://ggplot2.tidyverse.org/ |

## EXPERIMENTAL MODEL AND STUDY PARTICIPANTS

### Sepsis Immunomics cohort

The Sepsis Immunomics (SI) study is an observational study set up in Oxford to better understand the host response during sepsis via the use of immunological and high throughput omic approaches (South Central Oxford REC C, reference:19/SC/0296). Patients were eligible to join the SI study if they were ≥ 18 years of age and admitted to hospital with suspected sepsis at Oxford University Hospitals NHS Foundation Trust, UK. Recruitment was conducted at the intensive care unit (ICU), emergency department (ED), and hospital wards. In the ICU, patients were recruited if they had symptoms and signs of established sepsis (i.e. suspected infection and an acute change in sequential organ failure assessment (SOFA) score ≥ 2). In the ED and medical wards, patients were deemed eligible if they had suspected infection and a change in quick SOFA score (qSOFA) ≥ 2, as well as a National Early Warning System (NEWS2) score ≥ 7. Patients were excluded if consent could not be obtained; if there was an advanced directive to withhold life-sustaining treatment; when admitted to hospital for palliative care only; if pregnant or within 6 weeks post-partum; if presenting with

immunodeficiency due to steroid therapy, HIV infection or other immunosuppressive agents; and if diagnosed with metastatic disease or haematological malignancy.

Acute samples were collected at admission and repeatedly at regular intervals within the first 9 days of hospitalisation, most often at days 3 and 5 post-admission. A detailed list of the clinical variables referred to throughout this study, alongside their description and units of measurement is provided in Table S1. Patients contributing data to the analyses in this publication were recruited between 2022 and 2024.

### Healthy volunteer cohort
Volunteers self-reporting as healthy (HV) and with no recent history of infection were recruited at the Centre for Human Genetics, University of Oxford, following informed consent (ethical approval South Central Oxford REC B, reference 06/Q1605/55).

### METHOD DETAILS

### Collection and processing of plasma samples
Blood samples were collected in EDTA vacutainer tubes (BD Biosciences) and processed within 1 hour of collection. For plasma isolation, 10 ml of blood were centrifuged at 1,600 g for 10 minutes at 4°C. Plasma was obtained from the top layer following centrifugation and clarified by performing a second, high-speed centrifugation at 16,000 g for 10 minutes at 4°C to remove remaining cellular debris. Plasma was then aliquoted into 1.5 ml microtubes and stored at -80°C until DNA isolation.

### Collection and processing of whole blood leukocytes and CD66[+] cells
In a subset of individuals, whole blood leukocytes (WBLs) were obtained from EDTA-stabilised blood by separating the buffy coat layer after centrifugation at 1,600g for 10 minutes. In some individuals, neutrophil enrichment was also performed using the EasySep Whole Blood CD66b+ Positive Selection Kit (StemCell Technologies) according to the manufacturer's instructions. WBLs and CD66b+ cells were stored at -80°C until DNA isolation.

### Cell-free DNA isolation from plasma
Circulating cell-free DNA was isolated from 1 to 3 ml of high-grade plasma, as determined by material availability for any given patient. Isolation was performed using the QIAamp Circulating Nucleic Acid kit (Qiagen) according to the manufacturer's instructions. Carrier RNA was not added to avoid reads being taken up by the additional RNA during sequencing. DNA was eluted in 35 μl of elution buffer (Buffer EB; Qiagen) and stored at -20°C until library preparation.

Cell-free DNA concentration was estimated fluorometrically using the Qubit High Sensitivity dsDNA assay (ThermoFisher Scientific). DNA quality was assessed using capillary electrophoresis with the TapeStation 4200 system (Agilent Technologies) in combination with the cell-free DNA ScreenTape kit (Agilent Technologies). This enabled estimation of cfDNA purity based on the distribution of fragment sizes. Only samples with cfDNA purity $\geq$ 75% were taken forward for library preparation.

### DNA isolation from whole blood leukocytes
Whole blood leukocyte genomic DNA (gDNA) was extracted from 100 μl of frozen buffy coat or CD66b+ enriched cells using the NEB Monarch Genomic DNA Purification kit (New England Biolabs) according to the manufacturer's instructions. DNA concentration was estimated using the Qubit High Sensitivity dsDNA assay (ThermoFisher Scientific), and DNA quality was inspected with a TapeStation 4200 system (Agilent Technologies) and the genomic DNA ScreenTape kit (Agilent Technologies).

### Preparation of TAPS spike-in controls
Carrier DNA for TAPS library preparation was derived from PCR amplification of the pNIC28Bsa4 plasmid (Addgene) (reaction composition: 1 ng template DNA, 0.5 μM forward and reverse primers, 1X Phusion High-Fidelity PCR Master Mix (ThermoFisher Scientific) in HF buffer).

Positive and negative spike-in controls (CpG-methylated lambda DNA and 2-kb unmethylated DNA, respectively) were fragmented in a Covaris M220 ultrasonicator (Covaris, LLC). Size selection was then performed using a 1.2X ratio of AmpureXP beads (Beckmann Coulter), with the final median DNA fragment being 200 bp.

### TAPS library preparation
Between 10 and 50 ng or cfDNA were spiked-in with 0.15% CpG-methylated lambda DNA and 0.015% unmodified 2-kb control. DNA was next end-repaired and A-tailed using the NEBNext Ultra II End Repair/dA-Tailing Module (New England Biolabs). Ligation to Illumina Multiplexing adapters was then performed using the KAPA HyperPrep kit (Roche Sequencing Solutions) according to the manufacturer's instructions. Subsequently, ligated libraries were subjected to two rounds of oxidation with mTet1CD and reduction with pyridine borane (PyBr). In brief, DNA was oxidized by incubating it with mTet1CD in a 50 μl reaction for 80 minutes at 37°C (reaction composition: 50 mM Hepes buffer, 100 μM ammonium iron sulfate, 1 mM α-ketoglutarate, 2 mM ascorbic acid, 2 mM dithiothreitol, 100 mM NaCl, 1.2 mM adenosine triphosphate, and 4 μ Tet1CD). Next, 0.8 IU of proteinase K (New England Biolabs) were added, and the reaction was incubated for 1 hour at 50°C. Reaction products were cleaned-up using a Bio-spin P-30 Gel Column

(BioRad) and a 1.8X ratio of AMPure XP beads (Beckmann Coulter). Finally, Oxidized DNA was reduced by incubating it with PyBr in a 50 μl reaction for 16 hours at 37°C and 850 rpm agitation (reaction composition: 35 μl nuclease-free water, 600 mM sodium acetate solution, 1 M PyBr). Reaction products were purified using Zymo-Spin columns (ZymoResearch).

The final reaction products were PCR-amplified using the KAPA HiFi HotStart Uracil+ ReadyMix PCR kit (Roche Sequencing Solutions) using a pair of custom-made IDT primers (FW: 5'-AATGATACGGCGACCACCGAGATCTACAC-3', RV: 5'-CAAGCAGAAGACGGCATACGAGAT-3'). Amplification comprised four PCR cycles, after which DNA was cleaned up using a 1X ratio of AMPure XP beads (Beckmann Coulter).

### Targeted EM-seq library preparation

Enzymatic methyl-seq (EM-seq) was performed on a subset of samples for which WBL- or neutrophil-derived gDNA was available. To this end, 200 ng of gDNA were sheared using a Covaris M220 ultrasonicator (Covaris, LLC), producing a median fragment size of 200 bp. The NEBNext Enzymatic Methyl-seq Methylation Library Preparation Kit (New England Biolabs) was then used for library preparation according to the manufacturer's instruction. In brief, DNA was end-repaired, ligated to sequencing adapters, and multiplexed using combinatorial indexing. Indexed DNA was next supplemented with positive and negative spike-in controls (fully methylated *E. coli* pUC19A DNA and unmethylated lambda phage DNA, respectively), and enzymatic methylation was performed on this DNA using a two-step protocol consisting of oxidation of methylated cytosines with TET2, followed by deamination of unprotected (i.e. unmethylated) cytosines with APOBEC. This resulted in methylated cytosines retaining their identity and unmethylated cytosines being converted to uracil and sequenced as thymidines.

Converted DNA was amplified for five PCR cycles (reaction conditions: initial denaturation at 98°C for 30 seconds; five cycles of 1) denaturation at 98°C for 10 seconds, annealing at 62°C for 30 seconds, 2) extension at 65°C for 60 seconds; and final extension at 65°C for 5 minutes).

Amplified DNA was then hybridised to a panel of probes for targeted enrichment using the Twist Target Enrichment Fast Hybridisation protocol (Twist Bioscience). Individual libraries were combined into 8-plex pools for hybridisation with streptavidin baits comprising the Twist Human Methylome panel (Twist Bioscience), covering approximately 4 million CpG sites spread over 123 Mb of the genome.

Following hybridisation, library pools were magnetically purified and DNA was further amplified using 8 cycles of PCR (reaction conditions: initial denaturation at 98°C for 45 seconds; eight cycles of 1) denaturation at 98°C for 15 seconds, annealing at 60°C for 30 seconds, 2) extension at 72°C for 60 seconds; and final extension at 72°C for 1 minute). Final library pools were adjusted to the same DNA concentration and taken forward for sequencing.

### DNA sequencing

TAPS (cfDNA) and EM-seq (WBL) DNA libraries were sequenced using a NovaSeq X (Illumina) and NovaSeq 6000 (Illumina) instrument, respectively. In both instances, 150 bp paired-end reads were used.

## QUANTIFICATION AND STATISTICAL ANALYSIS

### Read alignment and filtering of TAPS data

Sequencing reads were first merged into a single file (FASTQ) per sample. Sequencing adapter trimming was next performed with TrimGalore (v0.6.10) using default parameters. Reads were aligned to a reference containing the human genome sequence (GRCh38, NCBI) and the spike-in sequences used during TAPS library preparation. Read alignment was performed using the Burrows-Wheeler algorithm[106] as implemented in BWA MEM (v0.7.17). Aligned reads were filtered by mapping quality (MAPQ score ≥ 10) and sorted by genomic coordinate using SAMtools (v1.8).[2] Finally, PCR and optical duplicates were marked and discarded from further analyses using Picard's (v2.23) *MarkDuplicates* function.

Following alignment, mapped reads were used to identify potential sample swaps using Picard's (v2.23) *CrosscheckFingerprints* function.

### Methylation calling from TAPS data

Sequencing library construction comprises an end-repair step which fills in 5' overhangs and removes 3' overhangs from input DNA. This results in methylation events being missed when they are located within the overhang of a DNA fragment, a phenomenon referred to as methylation bias (*mbias*).[115] We accounted for this by removing any CpG calls derived from read ends. To do so, MethylDackel's (v0.6.1) *mbias* function was used to assess the distribution of methylation estimates along the position of sequencing reads. This enabled us to define a set of clipping parameters which minimised potential bias originating from the original top (OT) and bottom (OB) strands, respectively (OT R1 start = 5 bp, OT R1 end = 135 bp, OT R2 start = 5 bp, OT R2 end = 115 bp, OB R1 start = 20 bp, OB R1 end = 145 bp, OB R2 start = 35 bp, OB R2 end = 145 bp). MethylDackel (v0.6.1) was used to clip read ends according to these parameters and to call methylation events at all CpG sites detected in clipped reads. Only bases with a Phred score ≥ 5 were kept for this analysis. Methylation calls were outputted in bedGraph and methylKit[111] formats.

Methylation calls were next quality filtered. BEDtools (v2.31) was used to remove any CpG sites mapping to centromeres, gaps in the genome assembly, regions flagged as 'blacklisted' by ENCODE,[116] or regions with a high density of repeats (based on UCSC

Genome Browser's *repeatMasker*[117]). Because TAPS cannot distinguish methylation events from C>T single nucleotide polymorphisms (SNPs), CpG sites intersecting any common SNPs (i.e. MAF > 1%) reported in the SNP database (dbSNP; v155) were also removed.[118]

MethylDackel was designed for analysis of bisulfite sequencing (BS-seq), which relies on conversion of unmethylated cytosines, as opposed to the direct conversion of methylated cytosines performed by TAPS. Thus, methylation calls had to be flipped (i.e. their effect direction reversed). Call flipping was performed using the *awk* command.

Finally, bedGraph files were converted to bigWig format for visualisation in the Integrative Genomics Viewer (IGV) genome browser.[119] mehtylKit files were imported into R (v4.3.2) for analysis.

### CpG quality filtering and summarisation of TAPS methylation data

Genome-wide methylation calls were imported into R (v4.3.2), quality filtered, and pre-processed using methylKit (v1.28.0).[111] First, mCpG>T conversion rates were estimated as the proportion of CpGs detected as methylated in the positive spike-in controls. Only samples with a conversion rate $\geq$ 80% were kept for analysis. CpG sites were next filtered by visualising sequencing coverage along each chromosome within each sample to identify outlier regions with unusually high coverage. Only CpGs mapping to autosomes and with a coverage of $\leq$ 25 reads were kept. Read counts per CpG were next normalised for varying coverage, and a union set of CpGs detected in at least 3 samples per group (i.e. sepsis patients or healthy volunteers) was defined using methylkit.

Feature summarisation was performed using methylkit by averaging normalised methylated and unmethylated read counts and estimating methylation proportions across all CpGs located within a given genomic window. Only windows containing $\geq$ 5 CpGs were kept. Feature summarisation was performed at the level of: 1) 1-kb tiles spanning the whole genome, 2) promoter regions for all annotated genes reported in GENCODE (v43),[120] 3) the first intron of each gene reported in GENCODE (v43),[120] and 4) regions contained in Loyfer et al.'s methylation atlas.[32]

### Methylation exploratory analysis and batch correction (TAPS data)

Methylation proportions tiled at the promoter, intron, and 1-kb tile levels were used for exploratory analysis and data visualisation. In brief, standard deviations were estimated per tile across the cohort and tiles were ranked by variability. The overlap between the most/least variable tiles and known functional elements (i.e. CpGIs, CpGI shores, exons, introns, and promoters) was assessed using methylkit and visualised using ggplot2 (v2.3.4.4). Tiles ranked amongst the top 80% by variability were used as a basis for hierarchical clustering and visualisation of sample dendrograms, as well as for dimensionality reduction using principal component analysis (PCA).

Batch effects between library preparation batches were identified based on PCA visualisation, and batch regression was conducted on tiled methylation proportions using the ComBat algorithm[112] as implemented in the SVA package (v3.50). Tiles with methylation proportions estimates > 1 or < 0 after batch correction were manually set to 1 and 0, respectively.

Following batch correction, sample clustering and PCA were recomputed. Principal components were visually explored to dissect their relationship with known clinical and technical covariates using ggplot2.

### Variance partitioning analysis of TAPS data

Batch-corrected methylation proportions tiled at the promoter and intron levels were used to identify the main sources of variability contributing to the circulating methylome. To do so, the variancePartition (v1.32.5) R package[46] was used to model methylation proportions as a function of known clinical and technical covariates using a linear mixed model. The following equation was assumed during model fitting:

$$p(mCpG) \ = \ \beta_0 + \beta\,x + b_i\,z + \varepsilon_i$$

Where $\beta_0$ represents the residual term and $\varepsilon$ a random error term, while **x** and **z** stand for variables modelled as fixed and random effects, respectively. When fitting the model across all samples (i.e. both healthy controls and sepsis patients), age and cfDNA yield were modelled as fixed effects, while disease status, self-reported sex, and individual were modelled as random effects. When fitting the model to sepsis samples only, age, time since admission, neutrophil-to-lymphocyte ratio (NLR), peak ALT, peak CRP, and peak Creatinine were modelled as fixed effects, with vasopressor use, self-reported sex, and individual being modelled as random effects.

The estimated proportion of variance explained per variable for each tile was visualised using the variancePartition and ggplot2 packages.

### Differential methylation and pathway enrichment analyses of TAPS data

Batch-corrected methylation proportions tiled at the promoter and intron levels were used for differential methylation analysis with a moderated T-test, as implemented in limma's (v3.58.1) *eBayes* function.[113] Differentially methylated regions (DMRs) were identified by: 1) contrasting cases and controls, and 2) by modelling methylation proportins as a linear function of ALT values. Regions were deemed differentially methylated if they showed evidence of a statistical association at a false discovery rate (FDR) $\leq$ 0.05 following multiple testing correction with the Benjamini-Hochberg procedure.[121]

The genes corresponding to each promoter or intronic DMR were used as an input for pathway enrichment analysis with the gene set enrichment analysis (GSEA) algorithm[122] using the fGSEA package (v1.28).[114] Enrichment was estimated by comparing the input

gene list with a series of cell type-specific gene sets obtained from previous single-cell experiments and maintained by the molecular signatures database (MSigDB).[123] Gene sets were deemed enriched if they showed a non-zero normalised enrichment score (NES) at FDR ≤ 0.05 following multiple testing correction. NES and FDR-adjusted P value estimates were visualised using ggplot2.

### Deconvolution of cfDNA tissues of origin from TAPS data

We next used methylation profiles to estimate the contribution of different tissues to cfDNA. Methylome deconvolution was performed taking as a reference either: 1) the BS-seq tissue methylation atlas published by Loyfer et al.,[32] or 2) a combination of the methylation array-based cell atlases published by Salas et al.[33] and Moss et al.[6]

Deconvolution based on the BS-seq reference was performed by averaging batch-corrected methylation proportions across all CpG sites detected in each region in the atlas. This methylation matrix was used as an input for cell type proportion estimation with the robust partial correlations (RPC) method as implemented in EpiDISH[56] (v2.18.0). Deconvolution based on the array-based references was performed by matching CpG sites to their corresponding Illumina array probe IDs and subsetting the CpG set to those sites present in either Salas's or Moss's atlas. The resulting methylation matrix was used as an input for cell type proportion estimation with the RPC method in a hierarchical manner, as implemented in hierarchical EpiDISH (*hepidish*).[92] Because the Moss atlas comprises solid tissues, while the Salas atlas focuses on circulating immune cells, we supplemented Moss's tissue atlas with an additional column ("*immune cells*"), which was defined as a weighted average of Salas's atlas, with each cell type being weighted by its average proportion in whole blood. This data set served as a reference for the first deconvolution step. Cell proportions estimated for the "*immune cells*" category were subsequently broken down into cell types by performing a second RPC-based deconvolution step based on Salas's atlas. This resulted in a final set of cell proportion estimates covering all solid tissues and circulating cells.

Cell-free DNA composition estimates per sample were visualised using ggplot2, and their correlation with clinical and technical variables was assessed in R.

### Cellular turnover analysis

Results from methylome deconvolution were used to estimate cellular turnover rates. To do so, we estimated the approximate number of genomes contributed by a given cell type per ml of plasma. We refer to this as genome equivalents (GEQs/ml), in line with previous studies.[6] Genome equivalents were calculated as follows:

$$GEQs \Big/ ml \left[\frac{genomes}{ml\ plasma}\right] = p(cfDNA)_i \cdot cfDNA\ purity \cdot \left(\frac{cfDNA\ yield\left[\frac{ng\ DNA}{ml\ plasma}\right]}{0.006\left[\frac{ng\ DNA}{genome}\right]}\right)$$

Where **p(cfDNA)ᵢ** represents the proportion of cfDNA contributed by cell type **i, cfDNA purity** the proportion of DNA estimated to be true cfDNA based on its fragment size distribution, **cfDNA yield** the amount of DNA obtained per ml of plasma, and **0.006** a constant representing the average DNA contained by a typical cell. GEQs/ml are proportional to the number of cell deaths occurring within a cell type.

Cellular turnover was estimated by integrating cfDNA GEQs/ml with absolute cell counts from clinical blood counts (CBC) carried out for the same samples. Turnover proportions were defined as the ratio of cfDNA GEQs/ml to CBC-based cells/ml for each cell type.

### Analysis of cfDNA fragmentation patterns

Following alignment and quality filtering, the genomic coordinates of each sequencing read were extracted using SAMtools (v1.8). Read start and end positions were used to infer the length of the corresponding cfDNA fragment. Fragment coordinates were then imported into R and read counts were tabulated against fragment length using tidyverse (v2.0.0), resulting in estimates of the proportion of cfDNA molecules of different lengths. Fragment length distributions were visually inspected using ggplot2.

Fragmentation patterns were contrasted between sepsis cases and controls by comparing the proportion of fragments which fell within subnucleosomal (<120 bp), mononucleosomal (120-250 bp), dinucleosomal (251-420 bp), or polynucleosomal (>420 bp) size ranges. Proportions were compared between groups using T-tests.

Fragmentation indices (FI) were estimated in line with an approach introduced previously.[65] In brief, we defined fragmentation indices as the ratio of the peaks in the fragment size distribution corresponding to mononucleosomal and dinucleosomal fragments:

$$FI = \frac{count(cfDNA\ fragments\ with\ length = 167)}{count(cfDNA\ fragments\ with\ length = 334)}$$

FIs were visualised using ggplot2, compared between groups using T-tests, and tested for associations with continuous clinical variables using Pearson correlation tests.

### Analysis of cfDNA fragment end-motif frequencies

Fragment end-motifs were obtained following the approach introduced by Jiang et al.[3] In brief, read start and end coordinates were used to fetch the reference genome sequence at the genomic location of interest using UCSC's *twoBitToFa* function. The first and last 4 nucleotides of this sequence were extracted using pattern matching with *grep*. To account for 5' overhang filling and 3' overhang removal during library preparation, end-motifs were converted to the 5' -> 3' orientation. To do so, the first four nucleotides (i.e. those mapping to the beginning of the 5' -> 3' strand of the reference genome) were taken directly, while the last four nucleotides (i.e. those mapping to the end of the 5' -> 3' strand of the reference genome), were reversed in order and their complimentary base pair sequence was derived using character substitution with bash's *tr* command. This resulted in both fragment end-motifs being in the 5'->3' orientation, thus representing the true sequence at the end of the fragment overhangs before end repair.

End-motif sequences were imported into R and tabulated, which enabled estimation of their frequency in each sample. To interpret the observed frequencies, BBMap's exact k-mer frequency calculation algorithm was used to compute expected frequencies for all 4-mer combinations based on the human reference genome sequence. Expected and observed frequencies were compared using scatterplots in R. End-motif frequencies were further used for dimensionality reduction with PCA.

Differential end-motif usage was assessed by contrasting frequencies between sepsis cases and healthy controls using Wilcoxon Rank Sum tests, along with estimation of $log_2$-fold changes (LFCs) in frequency for each motif. Wilcoxon p values were corrected for multiple testing using Benjamini-Hochberg's procedure. Associations between end-motif usage and quantitative clinical variables were identified using Pearson Correlation tests, with correlation p values adjusted for multiple testing using Benjamini-Hochberg's procedure. Differential end-motif usage was defined as evidence of different end-motif frequencies at an FDR $\leq$ 0.05.

### Cell-free DNA-based nucleosome footprinting

Nucleosome footprinting based on cfDNA coverage was performed following the approach by Snyder et al.[73] Briefly, sequencing reads were first filtered based on length to only include information from mononucleosomal fragments (i.e. fragment size $\geq$ 120 bp and $\leq$ 200 bp). Next, we tiled gene-containing regions by creating a 120-bp sliding window, which was subsequently slid by 1 bp steps across a 5 kb region centred at the TSS of each gene in GENCODE (v43). The overlap between mononucleosomal fragments and each sliding window was next computed, with fragments fully spanning the window (i.e. with start and end points outside the window) being tabulated separately from non-fully spanning fragments (i.e. fragments with a start or end position within the window). Windowed protection scores (WPS) were estimated as proposed by Snyder et al.[73] Namely, WPS was defined as the number of fully spanning fragments minus the number of non-fully spanning fragments. WPS values were adjusted for sequencing coverage differences by subtracting the mean WPS value for the region, after which they were smoothened using a normally distributed kernel with bandwidth set to 30 bp.

### Analysis of nucleosome footprints for different gene sets

Cell type-specific and disease-specific gene sets were collated from the following studies: 1) the Genotype-Tissue Expression (GTEx) project (v10),[105] 2) the Genomic Advances in Sepsis (GAinS) study,[20,80] and 3) the MSigDB database.[123] GTEx median TPMs were used to define a set of '*inactive genes*' (i.e. genes with zero expression across all GTEx tissues) and a set of '*highly active genes*' (i.e. top 500 genes with largest average TPM values across tissues). Bulk RNA-seq data from the GAinS study were used to group genes into quartiles based on their mean expression level in whole blood of sepsis patients. Finally, MSigDB signatures were subset to cell type-specific gene sets from Aizarani et al.'s single-cell liver atlas.[83] Nucleosome footprints were assessed separately for each gene set of interest.

### Modelling of nucleosome phasing using a harmonic oscillator

We propose that nucleosome phasing downstream the TSS can be captured by the average adjusted WPS value at each position for a given gene set, and that it oscillates periodically as a function of genomic distance. We believe this behaviour is well described by the following model:

$$WPS(d) = e^{-\kappa d} + A \cdot e^{-\lambda d} \cdot \cos(\omega \cdot d - \varphi) + b$$

Where **d** represents distance from the TSS (in bp). The first term models the general decay in WPS signal over distance, with $\kappa$ being the exponential decay constant. In contrast, the second term models the periodic oscillation of the WPS signal, with $\omega$ being the angular frequency of oscillations (in radians) and $\varphi$ the cosine phase displacement (also in radians). The magnitude of the oscillation, **A,** decreases over genomic distance, with this decrease being modelled by a further exponential function with decay constant $\lambda$. Thus, the second term functions as a dampened harmonic oscillator. Finally, **b** stands for the residual term.

This model was fitted to the observed WPS values in a two-stage process to aid model convergence. Fitting was performed with non-linear least squares (NLS) analysis using R's *nls* function. The exponential decay term was fitted first, with the dampened harmonic oscillator term being subsequently fitted to the residuals from the first regression. Model fitting was performed separately for each combination of gene sets and samples, with coefficient estimates from each analysis being extracted for future comparisons. Only WPS observations in the region spanning −185 to 925 bp from the TSS were used for model fitting. These capture the full NFR around the TSS, along with the first five downstream nucleosomes.

Model fits were assessed by substituting the estimated coefficients into the model equation and using it to predict WPS as a function of genomic distance. Observed and predicted values were visually compared using ggplot2.

Estimated coefficients were compared between sepsis cases and controls using Wilcoxon Rank Sum tests. Associations between model coefficient estimates and quantitative clinical variables were inspected using Pearson Correlation coefficients. Correlations were subsequently statistically tested using a Pearson Correlation Test, with p values adjusted using Benjamini-Hochberg's procedure.[121]

### Analysis of microbial DNA in circulation

Taxonomic classification of non-human reads was conducted as reported in previous studies.[25] In brief, properly paired sequencing reads generated from TAPS and which did not map to the human genome were extracted from BAM files following trimming and alignment using SAMtools (v1.18).[107] Reads were then classified into microbial taxa using the kraken2 algorithm.[108] Read classification relied on a custom-built database assembled with the *kraken2-build* function, which contained all publicly available RefSeq bacterial, viral, archaean, and fungal genomes. Microbial reads were next counted and tabulated at the genus level for each sample in our cohort, only keeping taxa with at least 10 reads detected. Finally, microbial abundances were defined as the proportion of non-human reads assigned to each taxon.

### Adapter trimming, read alignment, and methylation calling of EM-seq data

Raw sequencing reads were processed using nf-core's methyl-seq pipeline (v2.3.0).[109] This pipeline performs adapter trimming, read mapping, and methylation calling. Briefly, sequencing adapters were trimmed using TrimGalore (v0.6.10) with default parameters. Reads were next aligned to a reference containing the human genome sequence (GRCh38, NCBI) and the spike-in sequences used during EM-seq library preparation. Read alignment was performed using Bismark (v0.22.3),[110] which is optimised for BS-seq data analysis and is thus compatible with EM-seq's chemistry. Bismark removes duplicate reads, aligns reads to the reference genome, clips read ends to account for methylation bias (see TAPS library preparation methods), and identifies methylation events at C residues covered by clipped reads. Clipping parameters were set to 12 bp from each end of sequencing reads.

Methylation events were called per strand across all cytosine contexts. As CHG and CHH methylation were minimal, only CpG sites were kept for further analysis. Ratios of converted to protected CpGs were subsequently calculated and reported as methylation proportions. Mirrored CpG sites on opposite strands were collapsed into a single analytical unit using Bismark's *coverage2cytosine* function.

### CpG quality filtering of EM-seq data

Collapsed CpG methylation calls were imported into R using methylKit (v1.28.0).[111] Only CpG sites with a coverage ≥ 10 reads were kept for analysis. Outlying CpG sites in the top 1-percentile by coverage were also removed. Sites were also excluded if they overlapped any common (i.e. MAF > 1%) C>T SNPs reported in dbSNP (v151).[118]

### Variance partitioning analysis of EM-seq data

Methylation proportions were used to identify the main sources of variability contributing to the whole blood methylome using variancePartition (v1.32.5).[46] Methylation proportions were modelled as a function of known clinical and technical covariates using a linear mixed model as detailed above (see TAPS data analysis methods). The following variables were included in the model: age, cellular composition, phenotype (i.e. acute sepsis vs controls), sex, and smoking status. Quantitative variables were modelled as fixed effects, with categorical variables modelled as random effects. Cellular composition was defined as the coordinate of the principal component most associated with CBC-measured blood cell proportions during exploratory analysis of the whole blood methylome.

### Deconvolution of whole blood leukocyte methylomes from EM-seq data

Deconvolution was performed on EM-seq data from WBL and neutrophil samples with the RPC method as implemented in EpiDISH (v2.18.0).[92] The cell type methylation atlas of circulating immune cells published by Salas et al.[33] was used as a reference for this analysis. Cell proportion estimates per sample were visualised using ggplot2, and their correlation with the cell proportions reported in each patient's complete blood cell counts (CBC) was assessed using Pearson Correlation tests.

