## [Document S2. Transparent peer review records for Cano-Gamez et al · Cell Genomics]

The circulating cell-free DNA landscape in sepsis is dominated by impaired liver clearance

Author list

Kiki Cano-Gamez, Patrick Maclean, Masato Inoue, Sakineh Hussain, Elisabeth Foss, Chloe Wainwright, Hanyu Qin, Stuart McKechnie, Chun-Xiao Song, Julian C. Knight

Summary

Initial submission: Received : February 11th 2025

Scientific editor: Judith Nicholson

First round of review: Number of reviewers: 2
Revision invited : March 10th 2025
Revision received : May 15th 2025

Second round of review: Number of reviewers: 2
Accepted : 16th July 2025

Data freely available: Yes

Code freely available: Yes

This transparent peer review record is not systematically proofread, type-set, or edited. Special characters, formatting, and equations may fail to render properly. Standard procedural text within the editor's letters has been deleted for the sake of brevity, but all official correspondence specific to the manuscript has been preserved.

Referees' reports, first round of review

Reviewer 1

1. The authors reported an association between circulating levels of alanine aminotransferase (ALT), C-reactive protein (CRP) and cfDNA metrics. Similar observations have recently been reported by Malki et al. "Analysis of a cell-free DNA-based cancer screening cohort links fragmentomic profiles, nuclease levels, and plasma DNA concentrations" *Genome Research* 2025; 35: 31-42. The authors should therefore discussed these previously published observations together with their own data.
2. The authors claimed that their work represented the 'first high-throughput multi-modal study of cfDNA during acute sepsis'. I think that this sentence, and similar ones elsewhere in the manuscript, should be toned down. For example, Wang et al had previously reported their experience with analysing microbial DNA and their fragmentomic profiles in septic patients (Wang et al. "Fragment ends of circulating microbial DNA as signatures for pathogen detection in sepsis" *Clinical Chemistry* 2023; 69: 189-201).
3. Even though the authors have emphasised the potential difficulty in analysing microbial DNA, the above-mentioned paper by Wang et al clearly shows that this can be achieved. Hence, I would like to see some microbial cfDNA sequence analysis from the authors' dataset. This would be informative because the microbial pathogens are the cause of the patients' sepsis.
4. The authors showed that cell-free DNA concentration increased gradually for patients requiring progressively more invasive forms of respiratory support. Is this correlation the cause or effect of the different forms of respiratory support?

Reviewer 2

Summary

In this study, the authors comprehensively study the "circulome" in sepsis patients and controls. They use DNA methylation and fragmentation patterns and end motif sequences and nucleosome positioning signatures to infer tissue of origin. They studied 86 plasma samples from 46 sepsis patients and 12 healthy

controls. The authors found that cfDNA concentrations were >40x higher in sepsis patients compared to controls. They next used a multi-modal approach to profile circulating cfDNA to infer tissue of origin and was also compared to measures of severity of illness. They found that while the absolute amount of cfDNA was much higher in sepsis patients, the composition of cfDNA did not differ between patients and healthy controls. They use statistical approaches comparing organ failures and methylation patterns to infer organ of origin of cfDNA. They found that most cfDNA was of immune cell and erythroid precursors with some contribution from hepatocytes and endothelial cells. The correlate high levels of cfDNA and highly fragmented cfDNA and infer that the high levels are a result of loss of hepatic clearance of cfDNA. Finally, they study nucleosome footprint as a marker of gene activity and show increased gene activity in sepsis patients. Overall this is a very interesting study with provocative findings. Experiments are well done and findings are confirmed using multiple approaches. I do have several suggestions to improve the manuscript.

Comments

1. The use of health controls is problematic and does not allow conclusion that the findings are sepsis-specific. Rather, their findings could just represent liver dysfunction during critical illness general rather than in sepsis specifically. This is somewhat mitigated by the "dose response" seen with increased severity of illness but should still be discussed as a limitation.
2. Figure 1A shows data from 6 healthy controls (6 individual data points on bar graph) and not from 12 as discussed in the methods. Can the authors please explain this. Along the same lines, different numbers of control subjects are reported in different parts of the manuscript.
3. In Figure 1B, the cfDNA in pts on no oxygen look different in the two panels. It seems they are the same data so they should be identical. Also, it's not clear what is different between supp O2 and nasal cannula.
4. Throughout the manuscript the authors should provide more detail on the clinical measures. As an example, in figure 1D the authors show MAP on the day of sampling. Is this MAP at a specific time of day? Lowest MAP? Average MAP? Please apply this same suggestion throughout the entire manuscript.
5. The authors posit that, "monitoring the levels of hepatocyte-derived cfDNA could be a promising strategy for prompt detection of liver dysfunction." This is a bit of an over-statement. They do show that hepatocyte-derived cfDNA is elevated along with clinical markers of liver dysfunction but they do not show

that cfDNA levels precede liver dysfunction. Further, AST, ALT, and Tbili are sensitive markers liver dysfunction so the clinical utility of a new marker of liver dysfunction is not clear.

6. Hepatic dysfunction is defined differently in different parts of the manuscript. For the total and methylation pattern analysis, the authors use AST/ALT. This seems reasonable as they are hypothesizing that it is hepatic cell death that releases hepatocyte cfDNA. However, in their heatmap analysis of DNA fragmentation and end motifs, they use SOFA score which is based on bilirubin and not AST/ALT. SOFA is clearly clinically relevant for mortality prediction but the rationale for use here is not provided. Do patients with high liver SOFA scores also have high AST/ALT?

7. The authors should include a Table 1 describing the clinical cohort and control subjects.

Authors' response to the first round of review

Reviewer #1

Cano-Gamez et al reported their experience with analysing cell-free DNA in septic patients. I have the following specific comments:

1. The authors reported an association between circulating levels of alanine aminotransferase (ALT), C-reactive protein (CRP) and cfDNA metrics. Similar observations have recently been reported by Malki et al. "Analysis of a cell-free DNA-based cancer screening cohort links fragmentomic profiles, nuclease levels, and plasma DNA concentrations" *Genome Research* 2025; 35: 31-42. The authors should therefore discuss these previously published observations together with their own data.

Thank you for pointing us to this study. We have reviewed the findings by Malki et al. and updated our **Discussion** section accordingly. This now includes a comparison between the associations of cfDNA levels with ALT in both studies. In addition, we found their end-motif analysis intriguing and have expanded our **Discussion** to highlight the contrasting observations shown by both studies. We believe this has greatly strengthened our manuscript and has helped better place our study into context.

The updates to our **Discussion** section are copied below:

[1] "...While the discovery of such a prominent role for liver-mediated clearance in shaping the cfDNA landscape is surprising, it is in line with recent *in vivo* studies, which show that therapeutic agents that block hepatic clearance can substantially increase the amount of DNA recovered from liquid biopsies. This therapeutic blockade of cfDNA clearance is reminiscent of our observations in sepsis. *Moreover, our observations agree with a recent study of cfDNA in a large cohort of healthy individuals, which reports a significant association between liver function tests (ALT) and cfDNA concentration in plasma. Taken together, this indicates that cfDNA levels in circulation are tightly regulated by the liver, both during homeostasis and in acute infection...*"

[2] "... We hypothesise that observed increase in cfDNA levels in the absence of compositional differences could only be explained by impaired hepatic clearance of DNA and nucleosomal particles during sepsis. *This is supported by increased cfDNA fragmentation and higher frequencies of DNASE1L3-associated end-motifs, both of which suggest a prolonged exposure of cfDNA to circulating nucleases which is directly proportional to known biomarkers of liver dysfunction. It is worth noting how this differs from a recent study of cfDNA in a healthy population, where authors reported that individuals with higher levels of circulating cfDNA had reduced fragmentation and lower frequencies of DNASE1L3 motifs. These contrasting observations suggest there are different molecular*

mechanisms of cfDNA accumulation operating in health and disease. While higher levels of cfDNA in healthy individuals are likely caused by reduced DNASE1L3 activity, sepsis is instead characterised by a blockade of hepatic clearance by Kupffer cells, with the activity of DNASE1L3 in circulation remaining unaltered. This illustrates the complexity of cfDNA biology and highlights the need for further studies on this topic."

2. The authors claimed that their work represented the 'first high-throughput multi-modal study of cfDNA during acute sepsis'. I think that this sentence, and similar ones elsewhere in the manuscript, should be toned down. For example, Wang et al had previously reported their experience with analysing microbial DNA and their fragmentomic profiles in septic patients (Wang et al. "Fragment ends of circulating microbial DNA as signatures for pathogen detection in sepsis" *Clinical Chemistry* 2023; 69: 189-201).

We acknowledge that, while few sequencing studies have focused on host-derived cfDNA during sepsis, the same is not true for microbial DNA in circulation, which is a large and active area of research. We recognise that our previous statements did not make this clear and have rephrased them to reflect the existing body of research on this topic, as well as non-sequencing-based studies of cfDNA during this illness. All reworded sentences are clearly indicated in purple in our revised manuscript.

All updated sections are copied below:

Abstract:

[1] "...In conclusion, we present a detailed high-throughput multi-modal study of cfDNA during sepsis, which will serve as a reference point for future studies on the role of this biomarker in critical illness."

Introduction:

[1] "...Previous studies have demonstrated that cfDNA is significantly increased during sepsis and have suggested this relates to disease severity. For example, cfDNA concentration in serum is associated with sepsis mortality and higher levels of circulating mitochondrial DNA (mtDNA) predict poor prognosis. Moreover, the contribution of microbes and infecting pathogens to the cfDNA pool has been extensively studied. However, the molecular characteristics of human-derived cfDNA during sepsis, including its tissues of origin of and how it may be cleared from the blood, are not known..."

Discussion:

[1] "...Here, we presented a high-throughput multi-modal study of cfDNA during acute sepsis, comprising an in-depth analysis of circulating methylomes, fragmentation landscapes, and nucleosome footprints in a cohort of 56 samples from 31 hospitalised sepsis patients and 7 healthy controls..."

3. Even though the authors have emphasised the potential difficulty in analysing microbial DNA, the above-mentioned paper by Wang et al clearly shows that this can be achieved. Hence, I would like to see some microbial cfDNA sequence analysis from the authors' dataset. This would be informative because the microbial pathogens are the cause of the patients' sepsis.

We thank the reviewer for encouraging us to explore this question, which is clearly clinically relevant. Our study was not designed with microbial DNA applications in mind, which limits our power to reliably detect infecting pathogens. For example, we did not include non-targeting controls (NTCs) to accurately rule out microbial contaminants, and we believe we are likely losing a large proportion of the microbial DNA in circulation during library preparation. However, a re-evaluation of recent literature on this topic, including the study by Wang et al. pointed out by the reviewer, made it clear that there are computational methods which could help us circumvent some of these limitations. Thus, we revisited our data with the aim of identifying and classifying non-human reads.

We assembled a data analysis pipeline (see <https://github.com/jknightlab/TAPS-pipeline/tree/main/metagenomic-analyses>) which retrieves high-quality, properly paired sequencing reads which do not map to the human genome. The pipeline then classifies these reads into microbial taxa using the *kraken2* algorithm¹ and a database built from all publicly available RefSeq bacterial, viral, archaean, and fungal genomes (see revised **Methods**). We then quantified microbial reads at the genus level for each sample in our cohort, only keeping taxa with at least 10 reads detected. Finally, we defined microbial abundances as the proportion of non-human reads assigned to each taxon.

As mentioned above, we did not have NTCs which could be used to rule out contamination. However, we reasoned that microbial genera detected in healthy volunteers were more likely to be contaminants compared with those observed in sepsis patients, which are undergoing a severe infection. Thus, we tested for differential microbial abundance between sepsis patients and healthy controls and asked if microbes with higher abundance in the control group were enriched in previously reported metagenomic contaminants^{2,3}. Our analysis revealed a clear enrichment, confirming that most microbes with increased abundance in healthy controls are indeed contaminants (**Figure R1A**). This increased our confidence that, in contrast, microbes detected in septic patients are more likely to represent true infecting pathogens.

We next focused exclusively on sepsis samples and asked if the microbial compositional landscape reflected any known clinical characteristics for these patients. Principal component analysis of microbial abundances revealed two main sources of variation (jointly accounting for 90.7% of the variance): PC1 separated a sample with unusually

high levels of *Sphingopyxis* DNA, while PC2 separated two samples with high levels of *Escherichia* DNA (**Figure R1B**). The patient with high *Shingopyxis* abundance showed no detectable growth in their blood cultures. While this genus has been reported in polluted water sources⁴, there is no evidence of it being hazardous to human health, suggesting it could come from sample contamination. In contrast, both patients with detectable *Escherichia* DNA in circulation showed positive blood cultures for *Escherichia coli*, demonstrating that cfDNA retains valuable information on true pathogenic agents.

Motivated by these results, we set out to separate true biological signals from the large background of genera with lower read counts. To do this in a principled way, we made use of a stringent outlier filter. We defined true microbial cfDNA signals as those with an abundance at least 3.5 standard deviations above the cohort mean (**Figure R1C**). Only 11 microbial genera passed this threshold, including the two samples with positive *E. coli* cultures, a sample with *Staphylococcus* DNA and positive cultures for *Staphylococcus aureus*, and two faecal peritonitis samples with detectable *Enterococcus* DNA. We next directly assessed whether the 11 microbial genera detected in cfDNA were also picked up by blood or bodily fluid microbiological cultures. We observed that 28.6% of these genera were confirmed by microbiological testing (**Figure R1D**). This proportion was significantly higher than that of background microbial genera, which were only seen in cultures 0.6% of the time. Importantly, it is well known that a large proportion of infecting pathogens do not grow in laboratory settings, with some sepsis patients having detectable microbial DNA and higher mortality risk, but no observed microbial growth⁵. This could explain why some of the cfDNA signals were not confirmed by cultures. For instance, both samples with detectable *Enterococcus* DNA came from the same patient sampled at different times. This patient presented with faecal peritonitis, a condition where the leakage of *Enterococci* from the gut is common. This suggests that the signal is reproducibly detected over time, unlikely to be a contaminant, and consistent with disease biology, even in the absence of positive microbiology. Further studies with a more suited experimental design and deeper microbiological characterisation are needed to reliably assess the value of cfDNA to detect pathogen-derived nucleic acids.

We believe this analysis has greatly strengthened our manuscript and thank the reviewer for their suggestion.

Figure R1. Cell-free DNA contains material from infecting pathogens. **A.** Volcano plot showing microbial genera which are differentially abundant between sepsis patients and healthy controls. Microbial abundance in circulation (X axis) and statistical evidence (Y axis) are shown, with each dot representing a microbial genus. Red dots indicate known contaminants reported in previous metagenomic studies. Marginal density plots indicate the distribution of known contaminant species. **B.** Principal component analysis was performed on a matrix of microbial abundances from sepsis patient samples. Each dot represents a sample, with colours highlighting microbiological results from blood cultures. Marginal plots show the contribution of different genera to each principal component. **C.** Histogram showing the distribution of microbial abundances in cfDNA. The dotted line indicates the position of an outlier filter, which we used as a threshold to separate true signals from unspecific background. **D.** Bar plot of abundances for the 11 genera passing our outlier filter threshold. Each bar represents a sample which tested positive for the genus in question, with abundance estimates shown on the X axis. **E.** Bar plot showing the proportion of microbial genera detected in cfDNA which are also detected in blood or bodily fluid cultures. These proportions were compared to those of the cfDNA background using a Fisher's exact test.

These observations have been incorporated into our revised manuscript **Results** ('Cell-free DNA contains information on infecting pathogens') and **Methods** ('Analysis of microbial DNA in circulation') sections.

All relevant manuscript updates are copied below:

Results:

[1] *"Previous research has shown that cfDNA comprises a small proportion of fragments of microbial origin. Microbial DNA in circulation is particularly promising in sepsis, where infection is the root cause, yet identification of the pathogen causing the infection is often challenging due to the limited sensitivity and time-consuming nature of traditional microbiological testing. It has been suggested that metagenomic analysis of circulating cfDNA could help circumvent this limitation, as well as enable simultaneous profiling of the pathogen and the host immune response. Motivated by this research, we asked whether cfDNA contained any information about infecting pathogens.*

*We retrieved sequencing reads which did not map to the human genome and classified them into microbial taxa using a database of all assembled bacterial, viral, archaean, and fungal genomes (**Methods**). We then quantified microbial reads at the genus level and defined microbial abundances as the proportion of non-human reads assigned to each taxon (**Methods**). Metagenomic analyses are known to be highly sensitive to contamination. However, we reasoned that while microbial genera detected in healthy volunteers were probably contaminants, those observed in sepsis patients were more likely to be disease relevant. Thus, we tested for differential microbial abundance between sepsis patients and healthy controls and asked if microbes with systematically higher abundance in the control group were enriched in previously reported metagenomic contaminants. Our analysis revealed a clear enrichment, confirming that microbes detected in healthy controls are more likely to reflect contamination (**Figure 7A**). This increased our confidence that, conversely, microbial DNA detected in septic patients may represent true infecting pathogens.*

*We next focused on sepsis samples and asked if the microbial compositional landscape reflected any known patient characteristics. Principal component analysis of microbial abundances revealed two main sources of variation: the first component separated a sample with unusually high levels of *Sphingopyxis* DNA, while the second component separated two samples with high levels of *Escherichia* DNA (**Figure 7B**). The patient with high *Sphingopyxis* abundance showed no detectable growth in blood cultures, and while this genus has been found in polluted water, there is little evidence of it being hazardous to human health. This suggests that this signal may reflect sample contamination. In contrast, both patients with detectable *Escherichia* DNA in circulation showed positive blood cultures for *Escherichia coli*, demonstrating that cfDNA retains valuable information on true pathogenic agents.*

We next set out to distinguish disease-relevant signals from a large background of microbial genera with lower read counts. To do this in a principled way, we applied a stringent outlier filter. We defined outlying microbial cfDNA signals as those with an abundance at least 3.5

standard deviations above the cohort mean (**Figure 7C**). Only 11 microbial genera passed this threshold, including the two samples with positive *E. coli* cultures, a sample with *Staphylococcus DNA* and positive cultures for *Staphylococcus aureus*, and two faecal peritonitis samples with detectable *Enterococcus DNA* (**Figure 7D**). These results point to cfDNA retaining disease relevant pathogen information.

Finally, we asked whether the 11 outlying microbial genera detected in cfDNA were also observed in blood or bodily fluid microbiological cultures. We found that 28.6% of these genera were confirmed by microbiological testing (**Figure 7D**). This proportion was significantly higher than that of background (i.e. non-outlying) microbial genera, which were only seen in cultures 0.6% of the time.

In summary, we demonstrated that cfDNA contains disease-relevant information about the infecting pathogens underlying sepsis. While this is encouraging, further studies with larger sample sizes and deeper microbiological characterisation will be needed to reliably assess the value of cfDNA in pathogen detection.”

Methods:

[1] “Taxonomic classification of non-human reads was conducted as reported in previous studies. In brief, properly paired sequencing reads generated from TAPS and which did not map to the human genome were extracted from BAM files following trimming and alignment using SAMtools (v1.18). Reads were then classified into microbial taxa using the kraken2 algorithm. Read classification relied on a custom-built database assembled with the kraken2-build function, which contained all publicly available RefSeq bacterial, viral, archaean, and fungal genomes. Microbial reads were next counted and tabulated at the genus level for each sample in our cohort, only keeping taxa with at least 10 reads detected. Finally, microbial abundances were defined as the proportion of non-human reads assigned to each taxon.”

Discussion:

[1]: “...Finally, while we were able to detect pathogen-derived DNA using our data, we acknowledge that our experimental protocols are not ideally suited to profile this material, which is often highly damaged. Techniques specifically developed for damaged DNA profiling (e.g. single-stranded library preparation methods) could help bridge this gap, as could systematic detection of metagenomic contaminants based on non-targeting controls.”

4. The authors showed that cell-free DNA concentration increased gradually for patients requiring progressively more invasive forms of respiratory support. Is this correlation the cause or effect of the different forms of respiratory support?

We believe respiratory failure/support is unlikely to be the cause of the observed increase in cfDNA. This is due to several reasons. Firstly, we observe a negligible contribution of lung-derived DNA to the cfDNA pool (**Figure 4A** and **Figure S3**). If respiratory failure or the invasive nature or mechanical ventilation were the cause of this

increase, one would expect to see a clear signal of cell death coming from the lung epithelium. Secondly, cfDNA concentration is also associated with other markers of illness severity such as mean arterial pressure (MAP) and hospital location/care level, which indicates that cfDNA levels may reflect illness severity more generally and not specifically lung function or lung damage. Finally, we observe compelling and much stronger associations with liver function tests for both cfDNA levels and fragmentation features. Given the known role of the liver in clearing DNA from the bloodstream, we find this a lot more likely to explain our observations and account for the remaining associations with severity.

Reviewer #2

Summary

In this study, the authors comprehensively study the "circulome" in sepsis patients and controls. They use DNA methylation and fragmentation patterns and end motif sequences and nucleosome positioning signatures to infer tissue of origin. They studied 86 plasma samples from 46 sepsis patients and 12 healthy controls. The authors found that cfDNA concentrations were >40x higher in sepsis patients compared to controls. They next used a multi-modal approach to profile circulating cfDNA to infer tissue of origin and compare to measures of severity of illness. They found that while the absolute amount of cfDNA was much higher in sepsis patients, the composition of cfDNA did not differ between patients and healthy controls. They use statistical approaches comparing organ failures and methylation patterns to infer organ of origin of cfDNA. They found that most cfDNA was of immune cell and erythroid precursors with some contribution from hepatocytes and endothelial cells. They correlate high levels of cfDNA and highly fragmented cfDNA and infer that the high levels are a result of loss of hepatic clearance of cfDNA. Finally, they study nucleosome footprints as a marker of gene activity and show increased gene activity in sepsis patients. Overall, this is a very interesting study with provocative findings. Experiments are well done, and findings are confirmed using multiple approaches. I do have several suggestions to improve the manuscript.

Comments

1. The use of healthy controls is problematic and does not allow conclusion that the findings are sepsis specific. Rather, their findings could just represent liver dysfunction during critical illness generally, rather than in sepsis specifically. This is somewhat mitigated by the "dose response" seen with increased severity of illness but should still be discussed as a limitation.

Thank you for pointing this out. We acknowledge that the limitations of our control group were not sufficiently discussed in our initial manuscript. We have expanded our

Discussion section to better account for this, recognising that non-infection/sterile critically ill patients (e.g. post-surgery, pancreatitis, or trauma) may represent better comparator groups for future studies.

All relevant manuscript updates are copied below:

Discussion:

[1] "...While our study greatly expands our understanding of how the cfDNA landscape fluctuates during sepsis, it is important to acknowledge its limitations. Firstly, our sample size is still limited, and thus we are constrained by statistical power to only detect large changes in the circulome. *Moreover, our control group is composed of healthy individuals, which makes it challenging to understand which cfDNA alterations are specific to sepsis and which may be driven by more generalised alterations due to hospitalisation and critical illness. Future studies with larger sample sizes and better suited control groups (e.g. critically ill patients without infection) will be instrumental to develop more specific diagnostic tools...*"

2. Figure 1A shows data from 6 healthy controls (6 individual data points on bar graph) and not from 12 as discussed in the methods. Can the authors please explain this. Along the same lines, different numbers of control subjects are reported in different parts of the manuscript.

Thank you for identifying this issue. The discrepancy in the number of controls reported in different sections of the study arises from the fact that we quantified cfDNA levels in 12 control samples, but only 7 of these were ultimately sequenced. We have edited our **Results** and **Methods** sections to emphasise this and make it clearer.

Regarding **Figure 1A**, we identified a fault in our codes, which meant only data points from sequenced control samples (n=7) were being shown in this plot. We have now corrected this mistake and included all 12 control samples where cfDNA concentration was measured (see **Figure R2A** and revised **Figure 1** in our manuscript). Importantly, our conclusions from this analysis remain unchanged.

3. In Figure 1B, the cfDNA in patients on no oxygen look different in the two panels. It seems they are the same data so they should be identical. Also, it's not clear what is different between supplemental O₂ and nasal cannula.

We thank the reviewer for flagging this issue. The discrepancy between plots was caused by a small number of patients who were on different levels of respiratory support at different points of their hospital stay. We had not properly taken this into account,

resulting in some samples moving groups between visualisations. We have now modified our analysis to group patients according to their highest (i.e. ‘peak’) level of respiratory support throughout hospitalisation. This has removed any discrepancies between figure panels, with conclusions and associations derived from this analysis remaining unchanged.

With regards to the types of respiratory support, we recognise that the different methods of oxygen delivery were not clearly explained in our initial manuscript. With “nasal cannula”, we referred to oxygen which was delivered directly into the nostrils at low pressure. In contrast, the “supplemental O₂” group referred to patients for whom oxygen was delivered via a face mask, which results in a slightly higher pressure. We previously reported an FiO₂ of 24% and 28% for patients on nasal cannula and face masks, respectively⁶. Upon re-evaluation, we do not believe this slight difference in oxygen delivery is sufficiently important to warrant separating the patients into distinct groups. Thus, these are now treated as a single unit (“Supplemental O₂”), which we believe makes the figure more interpretable (see **Figure R2** and **Figure 1** in our revised manuscript). Crucially, our conclusions from this analysis remain unchanged.

Figure R2. Cell-free DNA levels in plasma are associated with illness severity. **A.** cfDNA concentration in the plasma of sepsis patients and healthy controls (upper panel), and in sepsis patients within different hospital care settings (lower panel). P values were derived using a Wilcoxon Rank Sum test. Box plots represent median and IQRs cfDNA concentration. **B.** cfDNA concentration in sepsis patients stratified by level of respiratory support. Categories are ordered

by increasing invasiveness. P values were derived using a Wilcoxon Rank Sum test. Box plots represent median and IQRs cfDNA concentration.

4. Throughout the manuscript the authors should provide more detail on the clinical measures. As an example, in figure 1D the authors show MAP on the day of sampling. Is this MAP at a specific time of day? Lowest MAP? Average MAP? Please apply this same suggestion throughout the entire manuscript.

This is an important point. We have added more detail on the meaning of the clinical variables used throughout the manuscript in a supplementary table (**Table R1** and **Table S1** in the revised manuscript).

We defined clinical measures as follows:

Table R1. Definitions of clinical variables used in this study.

Clinical variable	Definition	Units
Albumin	Lowest albumin measured on the day of sampling.	g/l
ALT	Highest (i.e. 'peak') alanine aminotransferase (ALT) measured on the day of sampling.	IUL
AST	Highest (i.e. 'peak') aspartate aminotransferase (AST) measured on the day of sampling.	IUL
Bilirubin	Highest (i.e. 'peak') total bilirubin measured on the day of sampling.	μM
Blood cell counts	Highest cell counts per cell type group observed on the day of sampling.	1×10^6 cells/ μl
Creatinine	Highest (i.e. 'peak') creatinine measured on the day of sampling.	μM
CRP	Highest (i.e. 'peak') C-reactive protein (CRP) measured on the day of sampling.	mg/l
Haematocrit	Lowest haematocrit measured on the day of sampling.	%
Haemoglobin	Lowest haemoglobin measured on the day of sampling.	g/l
INR	International normalised ratio calculated from the highest (i.e. 'peak') prothrombin time (PT) measured on the day of sampling.	-
MAP	Lowest mean arterial pressure (MAP) measured on the day of sampling.	mmHg
NLR	Neutrophil-to-lymphocyte ratio (NLR) calculated from the highest blood cell counts obtained on the day of sampling.	-

We refer to this table in the following revised manuscript sections:

Methods:

[1] "Acute samples were collected at admission and repeatedly at regular intervals within the first 9 days of hospitalisation, most often at days 3 and 5 post-admission. *A detailed list of the clinical variables referred to throughout this study, alongside their description and units of measurement is provided in Table S1.* Patients contributing data to the analyses in this publication were recruited between 2022 and 2024."

5. The authors posit that, "monitoring the levels of hepatocyte-derived cfDNA could be a promising strategy for prompt detection of liver dysfunction." This is a bit of an over-statement. They do show that hepatocyte-derived cfDNA is elevated along with clinical markers of liver dysfunction, but they do not show that cfDNA levels precede liver dysfunction. Further, AST, ALT, and Tbili are sensitive markers of liver dysfunction, so the clinical utility of a new marker of liver dysfunction is not clear.

We agree that current liver function tests are sensitive, and that more research is needed to assess the utility of hepatocyte-derived cfDNA as a biomarker. Furthermore, we acknowledge we do not have enough time series data to show an increase in this variable precedes liver dysfunction. We have reworded our statements to reflect this limitation and expanded our **Discussion** section to include a more balanced evaluation of this issue. While recognising the limitations of such a biomarker, we also highlight its potential advantages. For example, that cfDNA has a shorter half-life than aminotransferases, potentially providing a more immediate picture of the state of the liver; that it is amenable to measurement by qPCR, which has a very low limit of detection; that future development in deconvolution techniques and reference panels could make it possible to pinpoint damage affecting specific cell types and anatomical locations within the liver; and that our results suggest multiple aspects of liver function are captured by a single biomarker (see our response to point 6).

All relevant manuscript updates are copied below:

Results:

[1] "...We then turned our attention to liver dysfunction. We confirmed that hepatocyte-derived cfDNA GEQs were significantly associated with clinical signs of liver dysfunction, including ALT, aspartate aminotransferase (AST), and the international normalised ratio (INR), a measure of prothrombin time, and hence the synthetic function of the liver (**Figure 4E**). *Thus, hepatocyte-derived cfDNA correlates with liver dysfunction.* Similarly, we found a significant association between MEP-derived cfDNA GEQs and clinical measures of erythroid

production (haematocrit and haemoglobin levels). *This implies that MEP-derived cfDNA correlates with hematopoietic re-wiring and fluctuations in red blood cell (RBC) production.*"

Discussion:

[1] *"...In addition, it is unclear how this information could be harnessed for biomarker discovery. Biomarkers of liver function based on cfDNA would benefit from several advantages, including: 1) a shorter half-life of cfDNA compared with aminotransferases, which could result in a more immediate picture of the state of the liver; 2) the fact cfDNA is amenable to measurement by qPCR, which has a very low limit of detection; 3) the possibility that improvement in deconvolution techniques and reference panels could make it possible to pinpoint damage affecting specific cell types and anatomical locations within the liver; and 5) multiple aspects of liver function being captured by a single test. However, it is currently unclear how these features change over time, whether they precede liver dysfunction, and how much value they would add to the already existing battery of sensitive liver function tests. Further studies of cfDNA temporal kinetics will be needed to clarify this..."*

6. Hepatic dysfunction is defined differently in different parts of the manuscript. For the total and methylation pattern analysis, the authors use AST/ALT. This seems reasonable as they are hypothesizing that it is hepatic cell death that releases hepatocyte cfDNA. However, in their heatmap analysis of DNA fragmentation and end motifs, they use SOFA score which is based on bilirubin and not AST/ALT. SOFA is clearly clinically relevant for mortality prediction but the rationale for use here is not provided. Do patients with high liver SOFA scores also have high AST/ALT?

We thank the reviewer for pointing this out. Upon reflection, we realise we did not thoroughly explore the relationship between different cfDNA features and liver function tests, as well as the correlation structure of these variables. Thus, we revisited our clinical and cfDNA data to better understand this.

We re-tested for associations between the frequency of end-motif sequences in cfDNA and liver function tests, including ALT, AST, and bilirubin (which is used to calculate liver SOFA scores). We confirmed that a variety of 5'CC motifs were positively correlated with bilirubin, while 5'A motifs were negatively correlated (**Figure R3A-B**). However, we were surprised to see that this association was not present for AST or ALT (i.e. no motifs passing the multiple testing correction). To identify the cause of this phenomenon, we revisited the associations between all cfDNA features (i.e. concentration in plasma, fragmentation index, and end motif frequencies) and liver function tests, expanding our analysis beyond aminotransferases to also include total bilirubin and the INR. This revealed a clear pattern, where both cfDNA concentration in plasma and cfDNA fragmentation indices correlated with ALT, AST, and NLR, but not with bilirubin (**Figure R4A**). This suggests that these variables reflect hepatocyte cell death (i.e. a hepatocellular pattern of liver injury) but not necessarily a dysfunction in bile production

(i.e. a cholestatic pattern of liver injury)⁷. The opposite was true for 5'CC end motifs, which were significantly associated with bilirubin and INR, but not with AST or ALT, likely indicating cholestatic injury.

Figure R3. Cell-free DNA fragmentation features reveal different types of liver dysfunction.

A. Volcano plot showing the association between cfDNA end-motif frequency and total bilirubin. Each dot represents a motif, with their correlation coefficient (X axis) and FDR-adjusted p value (Y axis) from Pearson correlation tests shown. Significantly positively and negatively correlated motifs are highlighted in red and blue, respectively. **B.** Heatmap of 5' cfDNA end-motif frequencies identified as significantly correlated with bilirubin in our cohort. Shades of colour represent end-motif frequencies, with marginal colour bars indicating the first nucleotide in the motif (vertical axis), as well as disease status, ALT, and bilirubin (vertical axis). Samples were grouped using hierarchical clustering of motif frequencies. A subset of motifs with appreciable differences in frequency are highlighted. **C.** cfDNA fragmentation indices (Y axis, in logarithmic scale) stratified by time since admission (X axis). Each dot represents a sample, with colours indicating disease status. Box plots show median and IQR values for each group. P values were estimated using a Kruskal-Wallis test. For sepsis samples, correlation coefficients and p values between fragmentation indices and time were computed using a Pearson correlation test and

are also shown. **D.** Correlation plot between all pairwise combinations of liver function tests and cfDNA features. Each square indicates results from a pairwise correlation test, with shades of colour indicating the estimated Pearson correlation coefficient. Variable names are shown, with colours indicating whether they were derived from liver function testing or from cfDNA sequencing. Squares were grouped using hierarchical clustering of correlation coefficients.

Figure R4. Relationship between cfDNA features and liver function tests. A. Relationship between time since admission (X axis) and measurements for different clinical liver function tests (Y axis). Each dot represents a sample, with blue lines and shaded regions indicating fits and confidence intervals from locally estimated scatterplot smoothing (LOESS). Correlation coefficients and p values were estimated using Pearson correlation tests. **B.** Correlation between different clinical liver function tests (X axis) and cfDNA features (Y axis), including cfDNA concentrations in plasma (top row), cfDNA fragmentation indices (middle row), and frequencies of 5'CC end motifs (bottom row). Each dot represents a sample, with blue lines and shaded regions indicating linear fits and confidence intervals. Correlation coefficients and p values were estimated using Pearson correlation tests.

Previous studies of liver dysfunction, including drug-induced liver injury (DILI)⁸, septic shock, and hypoxic hepatitis (HH)⁹, have reported that some patients can progress from an initially purely hepatocellular injury (often caused by impaired perfusion and

oxygenation in the liver) to a later cholestatic injury, with marked elevations in bilirubin. A study of critically ill patients with liver dysfunction estimates that approximately 36% of patients will follow this trajectory, with a corresponding increase in their mortality risk⁹. These patients show a median delay of 2 days between presenting with peak ALT and peak bilirubin. This suggested to us that our observations may be explained by different clinical trajectories of liver dysfunction, with some fragmentomic features (e.g. accumulation of specific end-motif sequences) taking a longer time to develop or relating to specific mechanisms of liver dysfunction. To assess if this was the case, we studied the temporal behaviour of liver function tests and cfDNA features in our cohort. We observed that ALT and AST tended to peak at day 1 post-ICU admission, while bilirubin values peaked at day 5 (**Figure R4B**). This late increase in bilirubin was only seen in a small subset of patients, in agreement with only a minority of critically ill patients progressing to cholestatic injury^{9,10}. Cell-free DNA fragmentation also showed temporal variations, with fragmentation indices peaking at day 1 post admission and gradually decreasing throughout hospitalisation (**Figure R3C**).

To understand how these observations relate to each other, we computed correlations between all pairwise variables combinations (**Figure R3D**). This revealed three patterns of variation: 1) variables which reflect hepatocellular injury (ALT, AST, total cfDNA concentration in plasma, percentage of hepatocyte-derived cfDNA, and cfDNA fragmentation indices), 2) variables which reflect cholestatic injury (bilirubin and the proportion of 5'CC end motifs in cfDNA), and 3) variables altered in both types of injury (INR).

These observations suggest that different features of cfDNA may relate to different mechanisms of liver dysfunction and may show different temporal profiles and kinetics. This is encouraging, as it reveals how multiple potentially orthogonal layers of information can be extracted from the same biomarker. However, it also highlights the complexity of cfDNA biology and our limited understanding of how this material is cleared and metabolised. Future studies will be needed to clarify how these variables relate to one another and to build risk models which leverage all layers of information contained in cfDNA.

We thank the reviewer for their suggestion. We believe this re-analysis has greatly strengthened our manuscript and put our findings in a different light.

We have included these new observations in our revised **Figure 5** and a new supplementary figure (**Figure S4**), as well as in our revised **Results** section '*Cell-free DNA fragmentation reveals impaired hepatic clearance during sepsis*'.

All relevant manuscript edits are copied below:

Results:

[1] "...We next assessed the impact of sepsis on the end-motif landscape. Principal component analysis of end-motif frequencies showed a clear separation between sepsis patients and controls (**Figure 5F**), indicative of sizeable differences in motif usage. To understand whether the observed differences in end-motif frequencies reflected fluctuations in hepatic clearance, we tested for associations between end-motif usage and liver function tests (**Methods**). We observed that 5'CC end-motifs, associated with DNASE1L3, were more frequent in sepsis patients, and that they positively correlated with bilirubin (**Figure 5G-H**). In contrast, 5'A motifs were observed at lower frequencies in sepsis patients and were negatively correlated with bilirubin (**Figure 5G-H**). While DNASE1L3 is present in the circulation, DFFB is a primarily intracellular nuclease known to play a key role in apoptosis^{63,64}. Thus, our observations are compatible with a model where cfDNA is originally derived from apoptosis, but where impaired hepatic clearance subsequently causes prolonged exposure of cfDNA to DNASE1L3 in the bloodstream.

While the correlation between cfDNA end motifs and bilirubin supports this hypothesis, we were surprised to find no associations between end-motif frequencies and AST or ALT, the two most widely used markers of hepatic injury. To better understand this, we re-tested for associations between all combinations of cfDNA features and liver function tests, expanding our analysis beyond aminotransferases to include total bilirubin and the INR. This revealed that both cfDNA concentration in plasma and FIs correlated with ALT, AST, and INR, but not with bilirubin (**Figure S4A**). The opposite was true for the end motif landscape (i.e. 5'CC end motifs), which was significantly associated with bilirubin and INR, but not with AST or ALT (**Figure S4A**). This suggests that some cfDNA features may reflect hepatocyte damage (i.e. a hepatocellular pattern of liver injury), while others may reflect a dysfunction in biliary system (i.e. a cholestatic pattern of liver injury).

Previous studies have reported that some critically ill patients progress from an initial hepatocellular injury (often caused by impaired perfusion) to a later cholestatic injury, marked by a stark elevation in bilirubin. These patients show a delay of about three days between peak ALT and peak bilirubin readings. Thus, we reasoned our observations could be driven by different clinical trajectories, with cfDNA features exhibiting complex temporal profiles. To test this, we assessed the behaviour of liver function tests and cfDNA features over time. Both ALT and AST peaked at day 1 post-ICU admission in our cohort (**Figure S4B**), matching the temporal trajectory of FIs (**Figure 5D**). In contrast, bilirubin peaked at day 5 (**Figure S4B**), with this late increase only driven by a small subset of patients.

To further explore this, we computed correlations between all pairwise combinations of features to systematically group them (**Figure 5I**). This revealed three groups: 1) variables which reflect hepatocellular injury (ALT, AST, cfDNA concentration, percentage of

hepatocyte-derived cfDNA, and FIs), 2) variables which reflect cholestatic injury (bilirubin and the proportion of 5'CC cfDNA end motifs), and 3) variables altered in both injury types (INR). This demonstrates how different features of cfDNA could capture different mechanisms of liver dysfunction. This is encouraging, as it showcases how multiple orthogonal layers of information could be extracted from the same biomarker.”

Discussion:

[1] “Our study also highlights the complexity of cfDNA features and the need to better understand their temporal kinetics. For example, while cfDNA fragmentation correlated with markers of hepatocyte death, end-motif frequencies correlated with markers of cholestatic injury. This demonstrates that cfDNA clearance is complex and intimately linked to liver health. Future studies focused on disentangling the causal relationships between these variables are sorely needed.”

7. The authors should include a Table 1 describing the clinical cohort and control subjects.

We thank the reviewer for this suggestion. We have now added a table summarising the clinical and demographic characteristics of all 40 participants (corresponding to 58 samples) included in our TAPS sequencing cohort (see Table R2 and Table1 in our revised manuscript).

Table R2. Clinical and demographic information for patients in our cohort.

	Healthy volunteers	Sepsis patients
Sample size		
Number of participants	7	33
Number of samples	7	51
Median samples per participant	1	1 (range: [1 - 4])
Age		
Mean	43	61
95% CI	(37, 48)	(58, 63)
Self-reported sex		
Male (%)	57	69
Female (%)	43	31
Sepsis source		
Community-acquired pneumonia (CAP)		7 (21%)
Urosepsis		6 (18%)
Foecal peritonitis (FP)		4 (12%)
Biliary sepsis		3 (9%)
Necrotising fasciitis		3 (9%)
Bacterial meningitis		2 (9%)

Abdominal sepsis	1 (3%)
Infective Colitis	1 (3%)
Infective endocarditis	1 (3%)
Leptospirosis	1 (3%)
Septic arthritis	1 (3%)
Unknown	3 (9%)
Participant location	
Emergency department (ED)	4 (12%)
General ward	2 (6%)
Intensive care unit (ICU)	27 (82%)

References

- Wood, D.E., Lu, J., and Langmead, B. (2019). Improved metagenomic analysis with Kraken 2. *Genome Biology*. 20, 257. <https://doi.org/10.1186/s13059-019-1891-0>.
- Salter, S.J., Cox, M.J., Turek, E.M., Calus, S.T., Cookson, W.O., Moffatt, M.F., Turner, P., Parkhill, J., Loman, N.J., and Walker, A.W. (2014). Reagent and laboratory contamination can critically impact sequence-based microbiome analyses. *BMC Biology*. 12, 87. <https://doi.org/10.1186/s12915-014-0087-z>.
- Wang, G., Lam, W.K.J., Ling, L., Ma, M.-J.L., Ramakrishnan, S., Chan, D.C.T., Lee, W.-S., Cheng, S.H., Chan, R.W.Y., Yu, S.C.Y., et al. (2023). Fragment Ends of Circulating Microbial DNA as Signatures for Pathogen Detection in Sepsis. *Clinical Chemistry*. 69, 189–201. <https://doi.org/10.1093/clinchem/hvac197>.
- Sharma, M., Khurana, H., Singh, D.N., and Negi, R.K. (2021). The genus *Sphingopyxis*: Systematics, ecology, and bioremediation potential - A review. *Journal of Environmental Management*. 280, 111744. <https://doi.org/10.1016/j.jenvman.2020.111744>.
- O'Dwyer, M.J., Starczewska, M.H., Schrenzel, J., Zacharowski, K., Ecker, D.J., Sampath, R., Brealey, D., Singer, M., Libert, N., Wilks, M., et al. (2017). The detection of microbial DNA but not cultured bacteria is associated with increased mortality in patients with suspected sepsis—a prospective multi-centre European observational study. *Clinical Microbiology and Infection*. 23, 208.e1-208.e6. <https://doi.org/10.1016/j.cmi.2016.11.010>.
- Ahern, D.J., Ai, Z., Ainsworth, M., Allan, C., Allcock, A., Angus, B., Ansari, M.A., Arancibia-Cárcamo, C.V., Aschenbrenner, D., Attar, M., et al. (2022). A blood atlas of COVID-19 defines hallmarks of disease severity and specificity. *Cell*. 185, 916-938.e58. <https://doi.org/10.1016/j.cell.2022.01.012>.
- Strnad, P., Tacke, F., Koch, A., and Trautwein, C. (2017). Liver — guardian, modifier and target of sepsis. *Nat Rev Gastroenterol Hepatol*. 14, 55–66. <https://doi.org/10.1038/nrgastro.2016.168>.
- Yang, R., Li, K., Zou, C., Wee, A., Liu, J., Liu, L., Li, M., Wu, T., Wang, Y., Ma, Z., et al. (2022). Alanine Aminotransferase and Bilirubin Dynamic Evolution Pattern as a Novel Model for the

Prediction of Acute Liver Failure in Drug-Induced Liver Injury. *Front Pharmacol.* 13. <https://doi.org/10.3389/fphar.2022.934467>.

9. Jäger, B., Drolz, A., Michl, B., Schellongowski, P., Bojic, A., Nikfardjam, M., Zauner, C., Heinz, G., Trauner, M., and Fuhrmann, V. (2012). Jaundice increases the rate of complications and one-year mortality in patients with hypoxic hepatitis. *Hepatology.* 56, 2297–2304. <https://doi.org/10.1002/hep.25896>.
 10. Wang, D., Yin, Y., and Yao, Y. (2014). Advances in sepsis-associated liver dysfunction. *Burns & Trauma.* 2, 2321-3868.132689. <https://doi.org/10.4103/2321-3868.132689>.
-

Referees' report, second round of review

Reviewer 1

The authors have extensively revised the manuscript and have addressed all of my previous concerns. The additional analyses have greatly enhanced the manuscript.

Reviewer 2

The authors did a nice job responding to my comments. Only one additional comment. I noticed a typo in figure 1 B middle panel says “espiratory support” and should say “respiratory support”

Authors' response to the second round of review

N/A